# Marine Ranching Construction and Management in East China Sea: Programs for Sustainable Fishery and Aquaculture

**Xijie Zhou [†], Xu Zhao [†], Shouyu Zhang * and Jun Lin**

College of Marine Ecology and Environment, Shanghai Ocean University, Shanghai 201306, China;
d140302030@st.shou.edu.cn (X.Z.); xzhao@shou.edu.cn (X.Z.); jlin@shou.edu.cn (J.L.)

\* Correspondence: syzhang@shou.edu.cn; Tel.: +86-21-61900336

† These authors contributed to the work equally and should be regarded as co-first authors.

**Abstract:** Marine ranching, which is considered a sustainable fishery mode that has advantages for the ecosystem approach to fishery, the ecosystem approach to aquaculture, and capture-based aquaculture, is rapidly growing in China. The development of marine ranching requires integrating different theoretical frameworks, methodological approaches for conceptual exploring, and models and management of ecosystem frameworks. We reviewed the definition of marine ranching, the history of marine ranching construction in China, and the techniques, principles, and cases of marine ranching construction and management in the East China Sea (ECS). We highlight four major developments in marine ranching in the ECS: (1) marine ranching site selection and design, (2) habitat restoration and construction technologies, (3) stock enhancement and the behavioral control of fishery resources, and (4) marine ranching management. We conclude that this step-wise procedure for marine ranching construction and management could have comprehensive benefits in terms of ecology, the economy, and society. Finally, a synthesis of the existing problems in ECS marine ranching construction, along with future challenges and directions, are outlined.

**Keywords:** marine ranching; East China Sea; site selection and design; habitat restoration and construction; stock enhancement and behavioral control; marine ranching management

## 1. Introduction

Global fishery resources are increasingly threatened by the anthropogenic domination of the natural ecosystem and the concomitant impacts that accelerate irreversible damage to the habitat, ecosystem functioning, biodiversity, and the traditional fishery and artificial-culture industries. These effects occur through environmental pollution, overfishing, and climate change [1,2]. However, with a growing human population that requires high-quality protein, the demand for fish products is increasing, while fishery production has since plateaued [3]. These changes raise fundamental questions: How can we promote the sustainable use and conservation of fishery resources in an economically and environmentally responsible manner, while ensuring the upgrade of traditional fishery and artificial-culture industries? To address the critical question i.e., balancing the priorities between growth and competition, industrial and artisanal fisheries and aquaculture have emerged. Therefore, a new research area called the blue growth initiative (BGI) has attracted attention from many fields of science [4,5]. In line with the Food and Agriculture Organization's (FAO) BGI framework, the BGI focuses on four components: capture fisheries, aquaculture, ecosystem services, and the trade and social protection of coastal communities. Five targets were proposed: (1) achieving sustainable fisheries, (2) reducing habitat degradation, (3) conserving biodiversity, (4) maximizing social-economic benefits, and (5) assessing ecosystem services [6,7].

Well before the surge of interest in BGIs, different methods were proposed to address the issue of the sustainable use of fishery resources [8]. The terms restocking, stock enhancement, culture-based fisheries, capture-based aquaculture, and sea ranching or marine ranching have often been applied to describe the management activities in coastal fisheries. Many countries, such as China, Japan, Korea, and Norway, have regarded marine ranching as a robust tool for fishery resources stock enhancement, conservation, and use [9–11]. In the Encyclopedia of Ocean Sciences, ocean ranching is defined as follows: "Ocean ranching is most often referred to as stock enhancement. It involves mass releases of juveniles which feed and grow on natural prey in the marine environment and which subsequently become recaptured and add biomass to the fishery" [12]. At the Third International Symposium for Stock Enhancement and Sea Ranching (ISSESR), marine ranching was defined as the release of cultured juveniles into marine and estuarine environments for subsequent growth and harvest, with no intention that the released animals should contribute to the spawning biomass (though this can occur) [13]. Mustafa defined sea ranching as "releasing juvenile specimens of species of fishery importance raised or reared in hatcheries and nurseries into the sea for subsequent harvest at the adult stage or manipulating fishery habitat to improve growth of the wild stocks" [14]. Grant et al. suggested sea ranching consists of releasing hatchery-reared individuals into the wild, but with the expectation that individuals are harvested before they reproduce or mate with wild individuals [15]. While Kitada points out sea ranching aims at harvesting all juveniles (as possible) released in the harvesting areas, which can be managed by fishers [16]. However, as there are huge differences in the state of the marine environment, fish stock, fishery products demand, technology level, and marine policy between different countries and regions, and as marine ranching is an important activity, marine ranching (or sea ranching or ocean ranching) is defined heterogeneously.

The major differences in the definition of marine ranching mainly exist between Asian countries and other countries (South American countries, European countries, and Oceanian countries). For example, in South American countries, European countries, and Oceanian countries, the expectation is that species for sea ranching are harvested before they reproduce or mate with wild individuals [11,15]. In contrast, Asian countries expect that species for sea ranching return as adults to natal rearing areas to spawn [16]. Furthermore, studies on marine ranching in South American countries, Oceanian countries, and European countries refer to the use of stock enhancement and restocking strategies (i.e., releasing hatchery seeds to improve or rebuild fishery stocks) [12]. Most cases have shown that beneficiaries of ranching programs fall into four categories: releasing hatchery seeds to improve the self-sustaining populations, releasing hatchery seeds to rebuild severely depleted fish stocks, natural habitat conservation to maintain the habitat function of stock enhancement, and artificial reef construction to create an artificial hard substrate for reef fish [17]. In contrast, besides being focused on developing effective hatchery, release, and field technologies for restocking and commercial wild restock yield management [13], East Asian countries have concentrated on a concept of aquaculture-based, artificial habitat-based marine ranching (e.g., buoyant raft and artificial reef) and rehabilitation-based marine ranching (e.g., seaweed bed and seagrass) based on advanced engineering, new materials and structures, which has become a hotspot worldwide since the late 2000s [16–18]. The different trends in marine ranching modes between European and Asian countries may be due to their different fishery industry structures. Asian countries have been producing more farmed fish than wild catch since 2008, and its aquaculture share in total production reached 54% in 2012, with Europe at 18% and other continents at less than 15% [8]. Generally, many ecologists in Asia think that several different trophic level fishery outcomes can be achieved from a given set of natural and artificial conditions [10–12,18]. Namely, aquaculture-based marine ranching is more anthropogenic, since there is a greater possibility of managing and controlling the aquatic environment, material flow, energy flow, and other inputs, such as veterinary medicines. Two classical eastern marine ranching examples are the fishery management of marine ranching in the Oita prefecture in the Japan sea area, and the integrated multi-trophic aquaculture (IMTA) in Sanggou Bay, China [19,20].

Although China has the biggest capture and mariculture yield in the world, Chinese fisheries and aquaculture have continued to experience gradual but accelerated transformation and upgrading [6]. The government of China, along with the FAO, have proposed the 13th Five-Year Plan for Economic and Social Development of the People's Republic of China (2016–2020), with the goals of reducing capture fishery yield, fostering development of the distant-water fleet, substantially reducing the growth rate of aquaculture production, and developing marine ranching as a major mode of regional comprehensive and sustainable development through the use of artificial reefs, restoration of natural habitat, seasonal closures, restocking enhancement, and releasing. Marine ranching in China refers to:

A sustainable fishery mode that facilitates the breeding and conservation of fishery resource and marine eco-environment improvement using various measures such as artificial reef, stock enhancement, and releasing, to construct or restore breeding, growth, forage, and shelter habitat for marine organisms with ecosystem perspectives and principles in certain sea areas [21].

The concept of marine ranching in China is not specifically incorporated in the definitions of fishery, aquaculture, or capture-based aquaculture (CBA) proposed by the FAO. Instead, it consists of all the five different components—fishery, aquaculture, CBA, habitat conservation, and ocean engineering—and marine ranching is considered as a new type of fishery mode and marine economy in China. As a result, marine ranching plays an important role in the stock enhancement of fishery resources and the restoration of the marine ecosystem in China.

The rapid growth of marine ranching sectors worldwide, and the interaction amongst capture fishery, aquaculture, CBA, habitat conservation, and ocean engineering activities with other social and economic sectors and natural resources users, require a responsible and integrated framework and technologies for marine ranching development with Chinese characteristics. Chinese scientists and the government have developed numerous technologies, products, and policies for marine ranching. We reviewed the history, marine ranching pilot construction, systematic frameworks, and technologies for marine ranching within the overall context of marine ranching in the East China Sea (ECS).

## 2. History of Marine Ranching Demonstration Region Construction in China

The hypothesis that humans can build artificial ranches in the ocean by planting and cultivating marine creatures proposed by Zeng and Mao in 1965 provided the original idea for the Chinese marine ranching concept [22]. In the 1970s, the Conference on China Society of Fishery Restoration and Scientific Symposium, held on June 20, 1978, in Tianjin, Zeng defined farming and ranching of the sea to "use artificial disturbance to create a favorable sea environment for growth and development of marine creatures, while modifying appropriate traits of marine creatures to improve their quality and yield" [23]. After the embryonic concept of marine ranching was proposed in 1978, scientists in China introduced and implemented the relative framework, concept, and technologies of marine ranching in the areas of marine environmental improvement, incorporation of fishery and aquaculture activity, and hatchery and breeding technologies. For example, Feng proposed the theory of artificial reef driving forces to construct artificial reefs for marine ranching [24], Mao suggested that environmental improvements should be fully considered [25], Huang suggested making use of fishery cultivation in marine ranching construction [26], and Wang suggested that fishery should follow the road of farming [27]. As a result, China began experiments with enhancement and release, as well as experiments involving artificial reef design and construction. Artificial release and artificial reefs received significant attention in the 1980s and 1990s owing to the successful experience in Japan. Since 1983, Ministry of Agriculture of People's Republic of China has released 28,700 artificial reef units (i.e., 89,000 stacked cubic meters) under the guidance of scientists [28]. Before 2006, the marine ranching framework was divided into two separate sectors: capture fishery (artificial reefs and release) and aquaculture (cultivated farming).

Since marine ranching is linked to capture fishery and aquaculture, and as they usually occur in the same sea area, scientists in China recognized that the concept and technologies of both

capture fishery and aquaculture could be integrated into the concept of marine ranching. As a result, as of the 21st century, marine ranching in China has become an integrated framework with rapid development in marine spatial planning, ecosystem approach fishery, and ecosystem approach aquaculture. Notably, the above trends have occasionally aligned with the substance of Program of action on the conservation of living aquatic resources of China, which was stipulated by the General Office of the State Council of the People's Republic of China in 2006 and was considered as a milestone for sustainable fishery and conserving national ecological safety in China [29]. From 2000 to 2016, China has invested more than 5 billion yuan in marine ranching construction and has built 42 national marine ranching demonstration areas, with more than 200 marine ranches in an area of 852.6 km$^2$, where it has released 60,940,000 stacked cubic meters of artificial reef units. By 2017, marine ranching had hugely contributed to the economy. For example, the direct economic and ecological benefits of marine ranching have reached 31.9 billion yuan and 60.4 billion yuan, respectively [30].

Another impetus for marine ranching development in China was the 13th Five-Year Plan for Economic and Social Development of the People's Republic of China (2016–2020), which treated marine ranching as the most promising system and tool to solve a series of systematically critical ecological, social, and economic problems, such as overfishing, habitat loss, environmental pollution, transformation of the traditional fishery industry, and low marine resource levels and use efficiency [31]. The 13th Five-Year Plan declared that, for marine ranching, it would prioritize the regionally representative, high value of ecosystem service functions, scientific management, and social-economic benefits of modern marine ranching in the coastal zone of China, the Yellow-Bohai Sea, the East China Sea, and the South China Sea. For marine ranching pilot construction, 178 national marine ranching demonstration areas will be constructed before 2020 (The total construction area is about 1000 km$^2$ area, and 330 km$^2$ of seagrass and seaweed bed will be restored or rehabilitated, and another 50 million stacked cubic meters artificial reef will be released). The conservative estimation of social–economic–environmental benefits would reach 15 billion yuan per year [32]. In summary, the Chinese marine ranching mode has become similar to the concept, target, and framework of the BGI. According to the 2016, 2017, and 2018 Chinese Central Document No. 1, accompanied by the rapid development of powerful and coherent marine ranching-related concepts and technologies, traditional fishery in China will accelerate its transformation, upgrading to a sustainably integrated and socio-economically sensitive management fishery mode, which will meet the goals listed in the outcome document "The Future We Want" and the 2030 agenda for sustainable development.

## 3. Marine Ranching Construction in ECS: Techniques and Case

The East China Sea (ECS) is part of the Pacific Ocean and covers an area of roughly 1,249,000 km$^2$. Amongst the four sea areas of China, the East China Sea is characterized by three features, noting that the East China Sea is the most productive sea area of China. (1) According to the China Fishery Statistical Yearbook 2017, the East China Sea had the highest marine capture yield at about 5176 thousand tons/year, accounting for 39.0% of total capture yield of China [33]. (2) The East China Sea has faced the most severe marine environmental situation; the Bulletin of China Marine Ecological Environments Status 2017 reported that polluted water (water quality below level 1) had reached 60,480 and 80,000 km$^2$ in the summer and autumn, respectively. The main reasons for the low water quality are the high concentrations of nitrogen and phosphorus. Several sea areas (such as the Yangtze Estuary, Hangzhou Bay and Laizhou Bay, and the coastal area of Zhejiang, Jiangsu province) in the ECS have been facing varying degrees of eutrophication. (3) The ECS has been under enormous ecological, social, and economic pressures, and many well-known fishing grounds in the ECS (such as largest fishing ground of China, the Zhoushan fishing ground and the Yangtze estuary, and Yushan, Wentai, Mindong fishing grounds) no longer yield the capture they once did due to overfishing and the degradation of typical habitats like estuaries, tidal flat wetlands, and seaweed beds. However, given the demand for fishery products from a growing population, the pressure on the ECS alone will substantially increase,

while millions of fisheries and aquaculture-relative stakeholders are seeking a transformation to more sustainable fishery management and more effective conservation.

Building on the targets identified in China's 13th Five-Year Plan for the Economic and Social Development of the People's Republic of China (2016–2020) and the three objective characteristics of the ECS mentioned above, marine ranching construction in the ECS involves six elements mentioned by Yang: goals, ownership, seeds, space, diets, and management. The six core fields of marine ranching are performance evaluation, behavior management, breeding domestication, habitat rebuilding, baits enhancement, and systematical management respectively [10]. For the marine ranching-related technologies, ecologists have developed various tools for different targets (Table 1). Given the variety of potential tools, Zhang and Zhou suggested a technical procedure that contained four major sections [18]: (1) development within the context of marine ranching sites selection and design of marine spatial planning and ecosystem perspectives and principles; (2) development within the context of habitat restoration and construction technologies of ecosystem approaches and artificial infrastructure; (3) development within the context of enhancement and behavioral control of fishery resources of artificial structures and acoustic conditioning; and (4) development within the context of marine ranching management of ecosystem-based models and real-time monitoring measures.

**Table 1.** Aims of and tools for marine ranching.

| Aims | Tools |
| --- | --- |
| Conserve or protect ecological structures to enhance stock and biodiversity | Ecosystem approaches to fishery (EAF), marine ranching spatial planning |
| Protect ecologically valuable habitats and restore degraded habitats | Conservation and restoration of typical habitats (e.g., seaweed bed, mangrove, seagrass, etc.) |
| Promote appropriate use of marine space | Integrated multi-trophic aquaculture (IMTA), multi-trophic and spatial planning based marine ranching |
| Conserve vulnerable or regionally extinct species | Special marine protected areas (SMPA), species enhancement, and releasing |
| Construct high biological productivity habitats for the sustainable use of fishery resources | Artificial structure construction (e.g., artificial reefs, cages, buoyant rafts, artificial seaweed, mangroves, seagrass habitats). |
| Avoid and resolve social-economic conflicts | Spatial planning public or stakeholder engagement and regulatory system |

### 3.1. Marine Ranching Sites Selection and Design

Marine ranching site selection and design are the first steps in marine ranching construction. The FAO pointed out that marine spatial planning (MSP) and ecosystem approaches are the two robust tools for BGI. The FAO has provided guidance on spatial planning to many countries, including aquaculture zoning and site selection from an ecosystem perspective [34]. MSP and ecosystem approaches have been widely used for marine spatial planning in the ECS when marine ranching occurs on common property, such as the sea or natural water bodies. In general, both the MSP and ecosystem approaches are not activity-specific; they depend on the specific objectives and elements of marine ranching mentioned above.

Adoption of MSP in the ECS follows the literature and practices on the appropriate frameworks. These frameworks have been successfully implemented, adopting a pre-existing set of procedures to suit the prevailing circumstances in the ECS. A step-by-step example is marine ranching MSP in the Zhoushan grounds (Figure 1). In 2011, Xu summarized the three major principles of marine ranching site selection in the ECS as: (1) clarifying the main purpose of site selection; (2) analyzing the main factors affecting selection (i.e., marine ecological environment, including abiotic and biotic factors; infrastructure basement factors, including habitat modification, stock enhancement factors, ecological conservation functions; and social-economic factors, including fishing efforts, marine spatial uses of different stakeholders, and price of aquatic products); and (3) assessing all the sites according to the practical situation of the region, followed by a comprehensive evaluation of the primary results [35]. In 2013, Xu and Zhang improved the marine ranching site selection framework using the analytic

hierarchy process (AHP) method to build an evaluation model, where they suggested that the Zhoushan grounds should prioritize construction of the Ma'an archipelago marine ranching and Dongji island marine ranching [36]. In 2018, Zeng and Zhang further optimized the decision-making framework using the multi-criteria decision method, which integrated expert systems, geographic information system (GIS) spatial analysis, and raster calculations for protective artificial reefs in the Ma'an archipelago area, which is the national marine ranching demonstration area in China (Figure 1) [37]. Another representative marine ranching spatial planning framework was proposed by Duan, who analyzed the land elements in marine ranching and discussed the marine ranching landscape elements while proposing a landscape (spatial) planning direction for marine ranching [38].

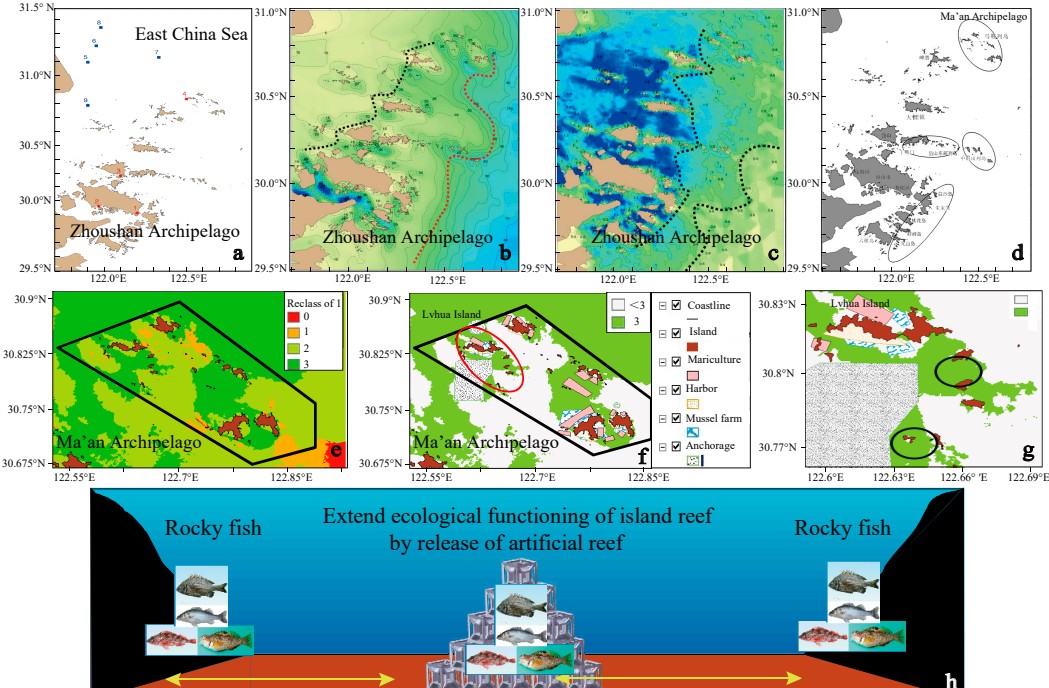

**Figure 1.** A step-by-step procedure for artificial reef construction site selection and design in the East China Sea (ECS): (**a**) obtain an approximate area for the proposed artificial reef; (**b**) determine depth distribution and the appropriate area for the artificial reef based on water depth; (**c**) determine turbidity distribution and the appropriate area for the artificial reef based on turbidity; (**d**) preliminary potential artificial reef site integrating depth, turbidity, and expert suggestions; (**e**) narrowing the field of potential construction areas where potential sites should be classified and graded using the analytic hierarchy process (AHP) method; (**f**) integrate the marine spatial uses of different stakeholders; (**g**) determine the appropriate construction area; and (**h**) develop the concept framework of the artificial reef-based island reef functioning enhancement technique.

A summary of the main MSP steps in the ECS is presented in Figure 2, although these steps are not prescriptive in a strict chronological order. However, the evidence suggests that the use of this framework has been successful in several marine ranching settings, so it can be recommended that the steps proposed in Figure 2 may be implemented as proposed to meet the objectives of marine ranching, at least in the ECS.

Marine ranching construction can have ecosystem effects [39,40]. Take artificial reef construction as an example: artificial reefs are human-made structures installed in aquatic habitats that serve as a substrate and/or shelter for organisms, create exclusion areas to reduce the effort of industrial fishing and are considered as an important method for ecosystem restoration [17,41,42]. Artificial reef construction may have both positive and negative environmental implications [43,44]. Although artificial reef construction can create suitable habitats for introduced (or target) species or

attracting fishes, such as rocky fish, the shift in the substrate from a soft substrate to a hard substrate may negatively impact non-target species and habitats [45,46]. When site selection and design is poorly executed, artificial reef construction can affect ecosystem functions and services with negative environmental, social, and economic consequences [47]. Also, for the reef construction, artificial reefs deployed in close geographical proximity to other reefs can provide additional habitat to support complementary communities of fishes [48]. Becker suggests that close module spacing supports a connected assemblage and Yang suggests that the shape, material, configuration and location of artificial reefs should be related with a specific goal to avoid mindless proliferation [49,50]. Therefore, marine ranching, as a sustainable fishery mode, demands construction suitability, sustainable practices at the level of the introduced (target) species, and taking responsibility for the interactions with local species (non-target species) in the ecosystem approaches context to minimize the negative consequence of blindly developing artificial reefs.

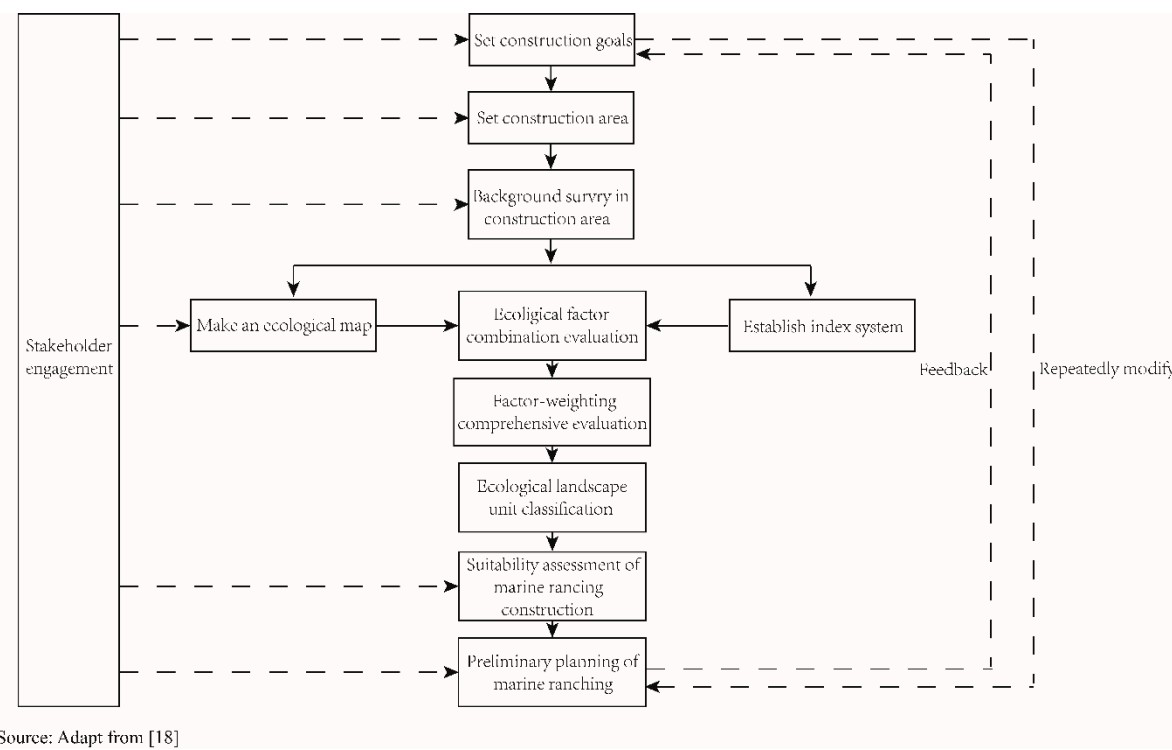

Source: Adapt from [18]

**Figure 2.** Procedures and linkages in marine spatial planning (MSP) implementation [18].

A conceptual diagram for the ecosystem approaches to marine ranching is shown in Figure 3 and the ecosystem approach to marine ranching has three objectives: (1) creating suitable habitats for the introduced (or target) species; (2) avoiding critical habitats of local (non-target) species; and (3) reducing competition between local (target) and introduced (non-target) species. Therefore, three key elements should be considered: (1) construction suitability, including functional zoning, basis of fisheries, planning, marine ecology environment, and infrastructure; (2) the introduced (or target) species, including seeds, breeding, domestication, release of introduced species, and the adaption of artificial habitats; and (3) local (non-target) species, including key habitats (spawning ground, nursery grounds, feeding grounds, overwinter grounds, and migration routes) of local species, and the adaption of local species to the artificial habitat. These approaches were applied in the Ma'an marine ranching pilot case [18].

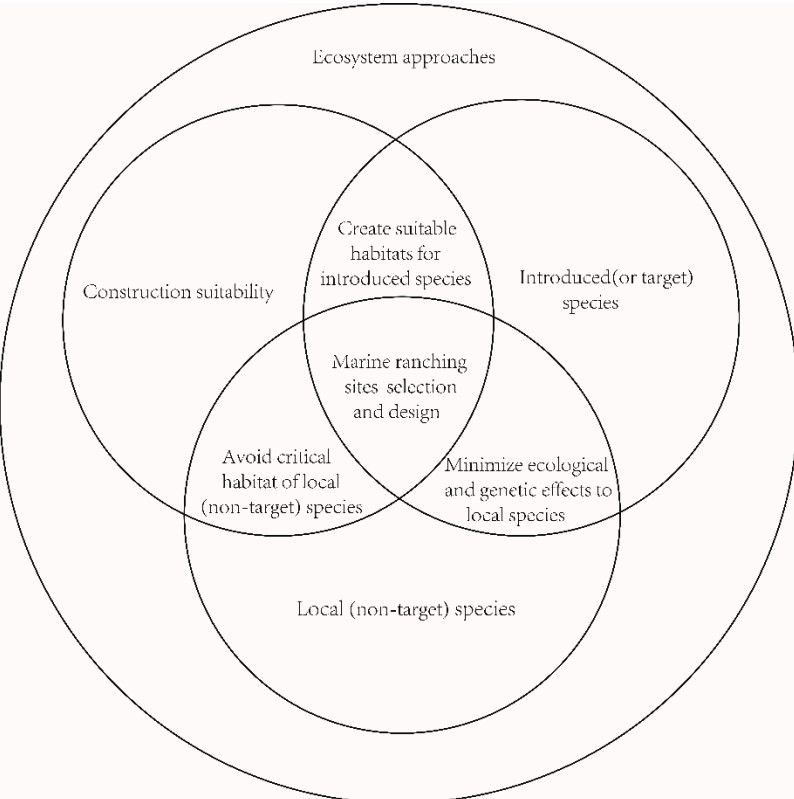

**Figure 3.** Conceptual diagram of the ecosystem approaches to marine ranching in the ECS.

## 3.2. Habitat Restoration and Construction Technologies

Marine ranching is underway in many ECS areas. To fulfil the aims of marine ranching, new marine ranching developments are ongoing or are being proposed. This section reviews the various techniques and issues to be considered in relation to habitat restoration and construction, inclusive of natural habitat (seaweed beds, mangroves, seagrass) conservation or restoration and artificial habitats construction (e.g., artificial reefs, cages, buoyant rafts, artificial seaweed, mangroves, and seagrass habitats) [50–53].

Conservation or restoration of natural habitats is an essential component of marine ranching in the ECS. Studies in the ECS have shown that natural habitats, such as seaweed beds, mangroves, and seagrass beds, have important ecosystem service functions. They are acknowledged as the most productive habitats and they provide abundant food sources for both the grazing food chain and detritus food chain [45,47,54]. They play important roles in maintaining biodiversity and have special importance in the life stages of species, e.g., as nursery grounds and feeding areas for migration species [55]. Natural habitats are hotspots for recreational fishery, cultural fishery, and livelihood fishery, and they have the potential to resist climate change by affecting carbon sequestration [56].

Following the considerable degradation of natural habitats, investments in habitat restoration, and construction have grown significantly in the ECS. China has implemented several national natural habitat conservation and construction initiatives, and these initiatives include three national programs: Program of action on the conservation of living aquatic resources of China, the National Marine Ranching Demonstration Areas Construction Program (NMRDA), and the 13th Five-Year Plan for the Economic and Social Development of the People's Republic of China, established in 2006, 2015, and 2016, respectively. For example, seaweed bed restoration has been conducted in the Nanji Archipelago Marine Protection Area (MPA), Ma'an Archipelago MPA and National Marine Ranching Demonstration Areas, Yushan Archipelago National Marine Ranching Demonstration Areas, and the Zhongjieshan Archipelago National Marine Ranching Demonstration Areas. Mangrove habitat

conservation and restoration have been completed in the Ximen Island National Special Marine Protection Area, and oyster reef habitat conservation has been conducted at the Shenhu Bay MPA.

Owing to special national conditions of China, China has the highest mariculture production yield in the world, where marine ranching construction in the ECS refers to the use of artificial structures [57]. Specifically, as shown in Figure 4, most typical marine ranching in the ECS has a strong relationship with aquaculture structures, such as artificial reefs, cages, and buoyancy cultures (commercial fish, seaweed, mollusks). These usages of artificial structures are similar to the concept of capture-based aquaculture (CBA), an integration of capture fishery and aquaculture proposed by the FAO [58]. In the ECS, marine ranching using artificial habitat construction technology, which has since been developed and commercialized, has reduced reliance on products from the wild for number of species and fishery resources.

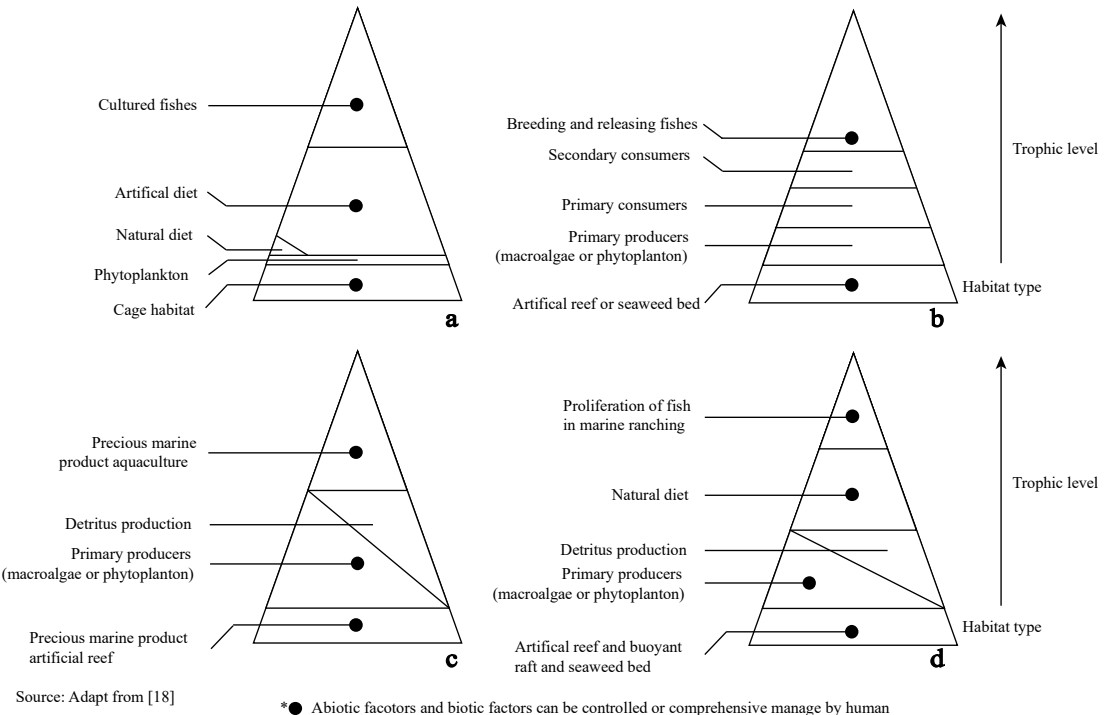

**Figure 4.** Typical marine ranching habitat construction mode in the ECS: (**a**) cage aquaculture mode, (**b**) traditional stock enhancement and releasing mode, (**c**) integrated multi-trophic aquaculture (IMTA) mode, and (**d**) multi-trophic and spatial based marine ranching mode [18].

### 3.2.1. Overall Restoration and Construction Mode Use in the ECS

Regarding the relative restoration and construction mode use in the ECS, we summarized the typical marine ranching construction mode into four types: (a) the cage aquaculture mode, (b) the traditional stock enhancement and releasing mode, (c) the integrated multi-trophic aquaculture mode, and (d) the multi-trophic and spatial based marine ranching mode. Appropriate consideration and selection of the restoration and construction mode for conservation or restoration of natural habitats and artificial habitat construction is essential to ensure that marine ranching is constructed in a way that guarantees over the long term, ecological and biological sustainability, economic efficiency, well-spatial-planning, and societal and animal welfare.

Cage aquaculture (Figure 4a) is a common method of aquaculture and CBA. In general, cage aquaculture is a highly artificially controlled habitat; specifically, the cage is highly dependent on artificial management and disturbance measures, such as artificial breeding, feeding, salinity, density control, disease control, waste water treatment, and harvesting [59]. Two steps have been taken to shift aquaculture to an ecosystem approach and to capture-based aquaculture. First, coastal marine ranching

uses cages as a polyculture structure in IMTA and multi-trophic- and spatial-based marine ranching modes, or as a hatchery structure for acoustic taming of juvenile fishes [60]. Second, offshore marine ranching uses larger and stronger cages for high-value carnivorous fish cultivation. However, despite using ecosystem approaches, the debates on the cage aquaculture method still exist on the following two aspects: (1) cage aquaculture continues to be fed using large quantities of low-value (trash) fish from the wild, often comprising juveniles from potentially valuable species [61,62]; (2) some fish cultured in cages occasionally escape into the natural environment as a consequence of human error, net breakages, or natural causes such as predator or storm damage to cages [63,64]. Species like salmonids are likely to have considerable invasive and genetic impact on local ecosystems [17,65–69]. To minimize the escape of cage aquaculture, acoustic conditioning can be used to recall escaped fish and when coupled with recapture, can reduce interactions with wild stocks and losses to the producer [70].

Traditional stock enhancement and releasing (Figure 4b) is the most common aquaculture type in the ECS. In general, stock enhancement is defined as the release of cultured juveniles into the wild population(s) to augment the natural supply of juveniles and to optimize harvests by overcoming recruitment limitations [71]. This mode consists of four phases: artificial breeding and propagation of release species, fostering before release, seeding release, and harvesting [13]. This method should follow the principles stated by Cheng and Jiang: (1) using a responsible approach to marine stock enhancement; (2) fully considering ecosystem perspectives, including ensuring that wild population adaption and genetic diversity will not be degraded and decreased after release of the artificial breed; (3) release of the species should not exceed the carrying capacity of the ecosystem; (4) maintain the structure, function, and balance of the ecosystem; and (5) ensure the comprehensive ecological–societal–economic benefits [72]. Once the seeding release in the sea is completed, the enhancement and release of recruitment depends on the abiotic and biotic factors of the ecosystem, including the natural and artificial habitats and the marine environment. In other words, the traditional stock enhancement and releasing method is highly dependent on natural and artificial habitat restoration and construction technologies, which is especially important for habitats with special significance to the life stages of species, e.g., as nursery grounds, migration routes, or feeding areas. Notably, compared to aquaculture mode, stock enhancement may have greater negative genetic effects on wild population, but the genetic effect diminished with the increased size of the wild population [73]. Given the complexity and uncontrollable risk of natural and artificial disturbances, there is still considerable uncertainty about the survival rate, recruitment, genetic effect, and quality during different life cycles of species release in the field and traditional enhancement and releasing cannot be successful over the long term unless sufficient efforts are also made to reduce harvest rates and rehabilitate natural habitats [73,74].

Traditional aquaculture is facing criticism for the use of large amounts of wild fish as feed and for environmental degradation. Ecological and environmental sustainability concerns have led to widespread interest in integrated multi-trophic aquaculture (IMTA; Figure 4c). IMTA is a polyculture of species with different trophic niches or functional groups (e.g., carnivorous fishes, such as finfish; primary production or inorganic extractive species, such as seaweeds; and organic extractive species, such as suspension and deposit-feeders). IMTA successfully develops a circulation aquaculture system that recaptures and converts some uneaten feed, wastes, nutrients, and by-products as valuable resources [60]. Various techniques have been developed for IMTA, such as integration of seaweed-fish culture, integration of mussels and oysters as biofilters in fish farming, and seaweed–cucumber–oyster–fish cultures, which has involved the combination of environmental, economic, and social aquaculture [75,76]. Most IMTA in the ECS open-water needs the strong support of marine engineering, especially artificial structures combined with aquaculture, e.g., seafood culture artificial reefs or suspended culture structures (i.e., seaweed, bivalve, or finfish buoyant raft longlines, or net cage farming). Therefore, besides marine engineering, five central issues should be fully considered during IMTA design: (1) water currents and hydrodynamic forces, (2) nutrient and energy accessibility, (3) fouling organisms and product quality, (4) temperature,

and (5) economic feasibility. Although some IMTA systems in China, such as the integration of seaweed, shellfish, and detritus feeders (cucumber) primarily in Zhangzidao, have been commercially successful at the industrial scale, IMTA still has a limitation for large-scale high trophic level cultures (especially finfish), since the trophic interactions between traditional cultured species (e.g., seaweed, filter-feeders, detritus feeders, and fishes) are not straight forward. The complex interactive processes that connect the biomass, nutrient uptake, and nutrient concentration of a balanced IMTA system can be difficult to fully examine under partially-balanced experiments or small-scale setups [58]. Usually, the integration of marine spatial planning, the creation of artificial reefs, and the restoration of natural habitats solves these limitations [18]. Meanwhile, although most species (e.g., scallop, abalone, sea cucumber, black sea bream, etc.) of large scale IMTA programs in China show no genetic effect on adjacent area, however, hatchery producers in China will need to give consideration to future potential genetic impacts of juvenile releases where wild stocks are present, especially for those species (e.g., salmonid, red sea bream, drum, steelhead, etc.) that may have negative genetic effects on the ecosystem [16].

Capture-based aquaculture (Figure 4d) was defined as the practice of capturing or collecting live material from the wild, and its subsequent direct use in aquaculture [77]. In many sea areas of the ECS, the construction and restoration mode would be better described as an ecosystem approach to integrated multi-trophic capture-based aquaculture (EAIMTCBA) rather than IMTA, which is closer to the integration of habitat restoration, IMTA, and stock enhancement. From ecosystem perspectives, appropriate marine ranching pilot construction in the ECS should fully consider the abiotic and biotic factors in ecosystem, and should incorporate operation of the ecosystem approach, which involves: (1) restoration of the natural habitat or building artificial structures, and (2) constructing biotic habitats through the artificial release of biota of different trophic levels into the artificial or natural habitats. This type of marine ranching is also described as the restore (or construct), put, grow, and take (or stock enhancement) operation. Also, as a systematic integration of habitat restoration, IMTA, and stock enhancement, a precautionary principle of both ecological and genetic effects is recommended particularly concerning animal health checks, potential genetic and invasive effects prior to release into EAIMTCBA systems.

### 3.2.2. Restoration of Natural Habitat or Building Artificial Structure

Arguably, degraded water quality, dredging, destructive fishing, climate change, and over exploitation caused by anthropogenic activities usually results in the modification of natural habitats, as well as losses in diversity, structural complexity, functional attributes, and ecosystem services [78,79]. As anthropogenic activities continue to increase, changes to the habitats and their inhabitants occur more frequently. For example, significant transitions have occurred from canopy habitats to turfs/barrens between 2000 and 2015 in Ma'an marine ranching. To stop the degradation trend, new natural or artificial habitats can intentionally or unintentionally be created [80]. The earlier view was dominated by the notion that natural reserves, rehabilitation or restoration were desirable because they would enhance the sustainability of habitats and the diversity of species. Kitada et al. suggest rehabilitation of the degraded natural habitats (e.g. nursery and spawning habitats) is the key to the success of marine ranching and stock enhancement [73]. To restore the natural habitat, transplanting macroalgae is possible but difficult, especially in areas that are wave-exposed or have high turbidity [81–83]. However, natural recovery can take decades due to slow growth rates of canopy-forming algae and it is usually limited to large-scale projects (usually taking place over 0–10 m in the ECS), leading to numerous attempts to actively add artificial habitats into the ecosystem while restoration of degraded reefs occurs [84]. In contrast, a popular view is that artificial habitats are part of the toolkit for managing the allocation and sustainability of harvestable resources [85]. The overall aim of purposely designed floating structures and artificial reefs is to create a hard substratum and novel habitat for flora and fauna. Chai et al. design a new artificial reef with pedestal and breeding broads and a procedure for macroalgae breeding was conducted in Gouqi Island, Zhejiang, China [86]. The speed at which the artificial

structures are colonized by plants and animals indicates that artificial habitats have the potential to extend ecosystem functions of natural habitats to offshore areas [86].

### 3.2.3. Constructing Biotic Habitats through Artificial Release of Biota of Different Trophic Levels in Artificial or Natural Habitats

Beyond the restoration of natural habitats or building artificial structures to optimize or improve ecological conditions at the pilot site, further efforts must be initially involved to construct biotic habitats by adding biota of different trophic levels in situ. These efforts should incorporate an explicit food-web perspective based on the bottom-up effect. As mentioned above, nutrient-rich pollution, known as eutrophication, is the most severe marine environmental problem in the ECS. Macroalgae aquaculture and seaweed bed restoration can remove nutrients from the environment, and as a consequence, the positive effect on seaweed growth from the high concentration of nitrogen and phosphorus may be more pronounced [87]. Macroalgae are the marine foundation species that demonstrate the following key qualities. First, they modify the physical environment by creating canopy- or turf-forming forests on natural habitats (rocky reefs) or artificial culture structures (buoyant rafts on surface layers and artificial reefs on the bottom layer) [88]. Second, they provide material resources (400 times net primary productivity (NPP) than phytoplankton), and food sources for both the grazing chain and detritus chain [89]. Although several studies have reported that the effects of macroalgae on higher trophic levels are indirectly mediated through food chain, previous studies have reported that macroalgae have often been postulated to contribute to habitat food webs via the grazing and detrital pathways, which is featured by rich prey species abundance and diversity [90]. Namely, primary production diversity has strong positive bottom-up effects on secondary producer species (the leakage between primary producers and high trophic level organisms) that graze primary producers directly or extract organic matter for growth [91]. Third, seaweed communities should serve macroalgae as nursery grounds due to them being an abundant food source, and Kitada et al. suggest rehabilitation of the degraded nursery habitat is indispensable to recover depleted populations and minimize the negative genetic effects of stock enhancement [73]. Generally, the success of marine ranching is highly dependent on the productive capacity of natural or artificial systems to support the release of commercial and recreational species. With flourishing and diverse food sources and the provision of physical structures, the release of fed fish (carnivore fishes) has the potential to contribute to sustainability [92,93].

Therefore, many marine ranching pilots in the ECS are designed based on the IMTA system through the propagation and planting of seaweed as well as estuarine release programs for stock enhancement [94]. Take the Ma'an marine ranching pilot construction as an example. The pilots were designed based on the ecosystem and engineering approaches to marine spatial planning at a 0 to 50 m depth (including artificial reef areas, seaweed bed conservation and restoration areas, and IMTA areas), which makes use of different marine structures and avoids conflict between the fishery community and stakeholders from other industries. Specifically, natural seaweed bed restoration occurred at 0 to 10 m, kelp and mussel cultivation occurred at 10 to 30 m, while artificial reef construction and fish-cage farming occurred at a 20 to 40 m depth characterized by strong currents. Since 2004, marine engineering in the pilot included about 1.16 million $m^3$ of artificial reef (including 1.1 million $m^3$ of artificial fish reefs and more than 50,000 $m^2$ of artificial seaweed reefs for habitat restoration), and about 14.7 million $m^2$ of IMTA structures (including fish-cage farming, long-line culture of kelp and mussels, lantern net culture of abalones, amongst others) in total. Additionally, 500 million artificially propagated seed or fish (including swimming crab, shrimp, mussels, jelly fish, black seabream, red snapper, grouper, yellow croaker, squid, etc.) were released or cultivated, while the yield of capture fishery and aquaculture from the Ma'an marine ranching reached 242.1 million tons and 170.9 million tons, respectively, in 2017 [18].

### 3.3. Stock Enhancement and Behavioral Control of Fishery Resources

The use of stock enhancement as a fishery management tool has renewed interest in marine ranching based on releasing hatchery-reared species and behavior control (or conditioning) [95].

However, several early attempts to enhance fishery stocks failed due to poor knowledge of the biology and ecology of species, the low fitness of hatchery-reared fishes due to behavioral and physiological deficits, and the unsuitable habitats and time of release [96–99]. In recent years, several successful results from stock enhancement have been reported globally, and marine ranching stocking effectiveness has been shown to be due to several aspects, especially the relationships between fish behaviors and habitats, fish size and release site, fish release and conditioning, fish quality, and release techniques [73, 100]. Efforts on behavior control (or conditioning) of fishery resources is another key to achieving sustainable fishery and aquaculture by minimizing the negative effects (both ecological and genetic) in the ECS. One way to follow the behavior control aspects is to use artificial structures related to the behavior of fish in marine ranching to design the operations. For those species with weak migratory behavior, algae and invertebrate community assemblages on artificial structures, and spatial structure complexity in artificial reefs, can differ from the assemblages in other habitats. As a result, artificial reefs and cultural structures can provide abundant food sources, suitable nursery structures to hide from predators, and aggregate fishery resources due to the behavioral preferences of species in many sea areas. Reef fish, such as black seabream (*Sparus macrocephalus*) and red seabream (*Chrysophrys major*), are two typical examples in the ECS that show that artificial structures can coincide with the forage and hiding behaviors of certain weak migratory species, respectively [101,102]. Several studies from around the world—such as studies on reef fishes associated with artificial reefs in the northwest Gulf of Mexico, gastropods in seaweed beds, and grunt fish (*Haemulon Sciurus* and *Haemulon flavolineatum*) in the backreef habitat—seem to be reaching similar conclusions [103–105]. The fish behavior and fish use of reef structure can be applied in the allocation of artificial reef, such as selecting minimum buffering zone between reefs to more effectively enhance fisheries by minimizing attraction of fishes from existing reefs, while also maximizing food resource availability for reef fishes and area for routine reef fish behaviors [106]. For species with moderate migratory ability, appropriate structures and habitats usually provide spawning, juvenile nursery, shelter, and forage functions during certain life stages of these species [107]. Therefore, by synthesizing species-specific behaviors and use modes of species within habitats, scientists have developed differently designed artificial substrata and reef structures [86,108,109]. Wu studied the reproductive and spawning behavior of cuttlefish (*Sepiella maindroni* de Rochebrune) and designed cuttlefish-specific spawning adhesion substrates. Similar species-behavior-specific stock enhancement cases can be found in the Zhongjieshan marine ranching pilot in the ECS [110]. Thus, for species with high migratory ability—such as the large yellow croaker, cod, and tuna—the application of offshore cages and life-cycle aquaculture allows the designing of optimal aquaculture schedules and embraces the growing demand of high-value fish [107,111,112]. Usually the cage should align with the species behavior, such as "milling", daily rhythms, preference for cage structure, use of space within the cage (pelagic or benthic nature), and diet behavior [113,114]. Fishery stakeholders have intensively promoted a series of new behavior-based aquaculture technologies and high yield-farming systems in the ECS, including offshore cages in coastal areas and deep-sea cage platform aquaculture in the open sea. However, the practical results from fish behavior tracking research have demonstrated that the artificial structures alone are not enough to: (1) attract the released juveniles to remain in the designated areas in the long term, and (2) prevent the escape of cultured fish from the cage. Additional behavioral studies have revealed that fish can be trained through social learning, which mostly depends on advanced technologies or methods adopted in the sea ranching operations using visual, auditory, or barrier cues [115]. Amongst the stock enhancement projects associated with behavior control in the ECS, the marine ranching of seabream (*Sparus macrocephalus*) in breeding is unique in that behavioral control is used instead of netting to keep the conditioned juveniles within hearing distance of the designed conditioning system even after months [81,116]. The proposed conditioning system in Xiangshan Bay marine ranch use: (1) mid-frequency sound (500–600 Hz) and light to call trained fish between feeding stations, where they received their food reward; (2) auto-bait casting systems to cast the bait quantitatively; (3) underwater video monitoring systems to monitor the effects of equipment, water

quality, and behavior of fish; and (4) communication systems, which included microwave spectrum or 4G, to control the fish acoustic condition equipment and to monitor the environmental factors and fish activity real time.

### 3.4. Marine Ranching Management

There is a growing desire to extend the tools on which marine ranch management is based to account for ecosystem considerations, and thereby achieve an ecosystem approach to marine ranching. Marine ranching management frameworks provide technologies to place fishery and aquaculture activities within the broader ecosystem context, ensuring that all stakeholders take full part in decision-making and in the implementation of appropriate measures and regulations. Rigorous monitoring of a large-scale marine stock enhancement program demonstrates the need for comprehensive management of fishery [73]; Zhang et al. suggested the core contents of marine ranching management should include systematic analysis, prediction, and optimization of complex ecological processes, and interactions in marine ranching [18]. Therefore, an actual marine ranch management framework can be created using three steps: conceptual explorations, strategic planning, and tactical decision-making [117]. Many studies, especially ecosystem models, focused on the application of ecological planning, an ecological approach to engineering, ecosystem assessment, and an ecosystem approach to decision-making [118]. Generally, the marine ranching management models can be categorized into three kinds of models: (1) conceptual models, (2) strategic models, and (3) tactical models. The type of model is determined by the management target and availability of data, whereas the selection of the model should reflect the problem being addressed [119].

#### 3.4.1. Marine Ranching Management Models

Conceptual models are aimed at developing an understanding of ecological processes in marine ranching and providing a base of information for strategic and tactical models. Conceptual exploration is based on ecological process-based rules suggested by Ogles in an increasing number of marine ranching pilots, whereas the ecological process explorations vary with the targets of marine ranching [120]. The models used for the illustrated interactions between parameters in the ecosystem using data collected from the system, laboratory, or field-based experiments, apply different model frameworks (e.g., generalized linear models, mixture models, Bayesian models, generalized additive models (GAM), etc.) to describe the relationships between parameters, and they use statistical diagnostic tools (e.g., residual analysis, variance estimation) to evaluate the models. For example, in a case study of the trophic interaction estimation between primary producers and macroinvertebrates in a seaweed bed in the ECS, laboratory-based dietary experiments, field exploration data, and stable isotope analysis were integrated, wherein a stable isotope linear model in a Bayesian- and process-based mixing framework was applied, and deviance information criterion (DIC) were used to evaluate the model performance [121]. Two additional examples are the quantification of relationships between environmental variables with community indices (e.g., fish richness and diversity indices) or meta-species indices in the Ma'an marine ranching pilots in the ECS. Principal component analysis (PCA)-based GAM and regular GAM were applied in both cases, and correlation analysis was chosen to compare the performance of the models [122,123]. The conceptual models also focused on the hydrodynamics, eutrophication levels, and single species habitat suitability index (HSI) [124–126]. Conceptual models are the basis of strategic and tactical models, and they are expected to provide mechanistic insight and predictive ability for a wide range of ecological problems [127].

Strategic models, which focus on a broad scale assessment of directions and patterns of change, have been gaining recognition as robust tools for long-term management and planning of comprehensive benefit due to the combination of biological, ecological, economic, and social factors [128]. Strategic models can be used to simulate long-range or broadly-based trade-offs of different stakeholders and policy objectives. Many approaches have been applied to test marine ranching sustainability in China. For example, Shi et al. suggested the IMTA of kelps and scallops

as the most sustainable culture model from the economic and ecological aspect, and used energy analysis to assess the ecological-economic benefits of monocultures and IMTA in the Sanggou Bay of China [17]. Zhao et al. successfully used the energy ecological footprint to compare and monitor the environmental impacts of fish farming in the ECS [129]. Xu et al. used the pressure–status–response model to evaluate the eutrophication levels in Xiang Shan Bay in the ECS [130]. Strategic management advice should be based on a combination of the different conceptual models. For example, Chao et al. proposed an integrated regional carrying capacity assessment indices system combined with a conceptual model of "driving force–pressure–state–response–control" (D-PSR-C) in the ECS for long-term sustainable development of the economy and ecology [131]. Sometimes different models can be used to answer the same strategy problem, and models providing strategic advice can be selected based upon the data, where a strategic model selection example was established by Zhang et al. (Figure 5) [18].

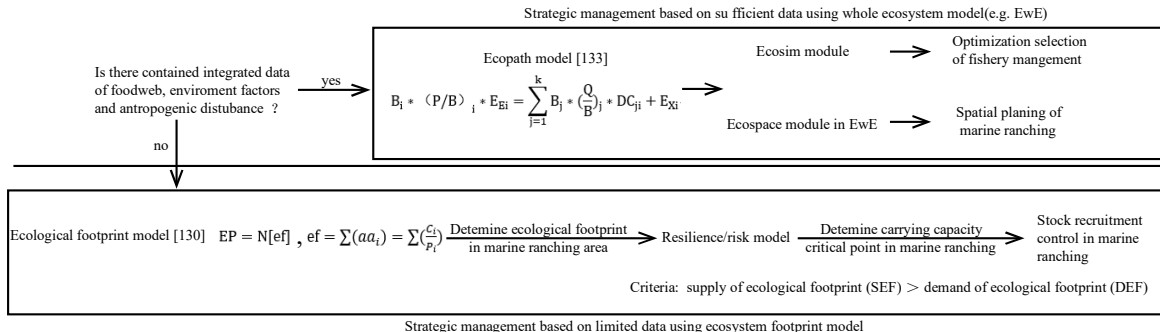

Source: Adapt from [18]

**Figure 5.** An example of strategic model selection based on sufficient and limited data using the Ecopath with Ecosim (EwE) model and the ecological footprint model, respectively. Note: in the ecopath model, $(P/B)_i$ is the ratio between production and biomass of group i, $(Q/B)_i$ is the ratio between consumption and biomass of group i, $DC_{ij}$ is the proportion predator i in total consumption of prey j, $EX_i$ is the production (including the catch and net outflow). In the ecological footprint method, i is the primary or secondary producer in marine ranching, $P_i$ is the primary or secondary production in marine ranching, $C_i$ is the introduced species i's average consumption biomass to primary or secondary production, $\alpha\alpha_i$ is the mean production area of the introduced species consumption to primary production i, N is the number of introduced species, ef is the mean ecological footprint of the introduced species, and EP is the total ecological footprint [18,129,132].

Tactical models, which are directed at supporting specific management decisions in the short-term (e.g., the next three to five years), are linked to an operational objective, and are in the form of a rigid set of instructions. These models have been rapidly developed worldwide. In China, although tactical models are rarely reported for making tactical decisions, some management practices in the ECS that been implemented as a continuum running from conceptual understanding and long-term strategic decision-making to tactical decision-making. An example involves a stepwise system project in marine ranching pilot management in the Ma'an archipelago in the ECS. Before marine ranching construction, fishery resources and habitats had experienced continued decline and degradation; as such, several management decisions have been developed based on continuous conceptual exploration, data collection, and modeling since 2003. Specifically, in the Ma'an marine ranching pilot, the seaweed bed was managed to provide a standard procedure for natural seaweed conservation and artificial seaweed bed construction and restoration based on conceptual models, such as primary productivity, nutrient recycling, static isotope food web structure, mass-balanced ecopath model, modeling predator–prey interactions, and the ecological functional boundary of the seaweed bed [119,133–135]. As a result, the seaweed habitat degradation trend has eased since 2011, and further efforts are aimed at constructing or restoring another five hectares of artificial seaweed bed before 2025 (i.e., 2020–2025). Secondly, a preliminary management procedure for artificial reef construction and stock enhancement has been adopted to provide seasonal closure, regional closure (nearshore and some artificial reef areas), hatchery-reared

juvenile release, fishery research sampling frequency fishing gear, and artificial reef management, so that the artificial reef can be appropriately allocated and constructed to improve the habitat suitability for certain commercial species [37,136–138]. Third, several preliminary strategic management procedures for mariculture have been conducted to ensure culture sustainability. For example, the allocation of mussels in artificial-mussel farms is based on hydrodynamic simulation models to accelerate nutrient cycling in the cultured area, where the designed artificial reef will be released before 2025 [124].

### 3.4.2. Real-Time Monitoring and Long-Range Data Communication

For marine ranching management, the choice to assess marine resources and their environments using quantified and habitat factors is ideal when creating appropriate ecosystem-based models and obtaining a wide range of information simultaneously via real-time monitoring and long-range data communication. Generally, real-time monitoring data—such as sea surface (or bottom) temperature, chlorophyll-a (chl-a), harmful algae abundance, sea surface height, nutrient salt, salinity, dissolved oxygen, and turbidity—are closely correlated with the enhancement of fishery resources, the evaluation of a real-time-monitoring-based ecosystem health index (EHI), and the development of automatic data processing [139,140]. Amongst the different methods available for real-time monitoring in the ECS (e.g., remote sensing, Argo global observation networks, aquaculture platform monitoring systems), each method has its own advantages and scale. Specifically, remote sensing is advantageous for the rapid and large-scale collection of data, but its defect is that it is limited to the few factors (e.g., sea surface temperature, chl-a, sea surface height turbidity, and surface salinity) that can be monitored [141,142]. Argo global observation networks have the advantage of having the potential for multi-sensor integration and data fusion to meet with the specific requirements of different stakeholders, especially marine hazards prediction and warning, whereas the disadvantages include high cost and relatively low resolution [143,144]. The aquaculture platform monitoring system has tremendous strength in terms of expansion and maintenance of the system, where it is applicable to more general regions than other methods, and can be installed in the special regions, such as artificial structures (e.g., artificial reefs, acoustic condition equipment, cages, etc.) [101,145–147]. The aquaculture platform monitoring system can incorporate more parameters compared to other methods and the data directly reflect the environment of the marine ranching area, though not effective enough for hazard forecasting. In summary, the different methods have their own merits and drawbacks, and the integration of monitoring methods combined with the Internet of Things (IoT) and the Internet+ framework are suggested as robust tools for tactical-decision-based management in the ECS [10,18].

### 3.4.3. Management of Complex and Multi-Stakeholder Users and Uses

Much of marine ranching areas are used for more than one activity, especially in coastal and archipelago areas, which sustain many anthropogenic activities. For the marine ranching area to function well and sustainably, multiple activities and marine uses need to operate equitably and successfully with the various combination of users and uses. An example of cooperative management would be the multi-stakeholder co-uses example in the Ma'an marine ranching pilot in the ECS. The uses of the marine ranching pilots include commercial fishing, marine transportation, offshore mariculture, carbon sequestration sites, ports and anchorage, recreational fishing, scuba diving cables, pipelines, transmission lines, and scientific research. Therefore, administration in the Ma'an marine ranching pilot must coordinate the different uses and stakeholders, such as the separation of aquaculture structures from shipping lanes to avoid collisions, or the separation of artificial reefs from bottom trawling to avoid damage to the equipment. To achieve high-level policy goals, such as sustainability of harvestable resources and transformation of the traditional fishery industry as mentioned in the 13th Five-Year Plan, all stakeholders in the Ma'an marine ranching pilot have set up several management frameworks [31]. First, the process for the development of a management plan follows two principles: to collect and use the best available information and to have broader stakeholder engagement and consultation (Figure 6). Second, a regulatory system was created that combines government supervision and law-enforcement,

involving stakeholder self-regulation, scientific internal communication, and control (Figure 7). Third, a tool and measures box was introduced to address the various challenges of managing the different components of marine ranching (Table 2). Together, these sets of frameworks provide the means to plan and manage marine ranching development that integrates it with broader stakeholders and considers the overall social, ecological, and economic benefits.

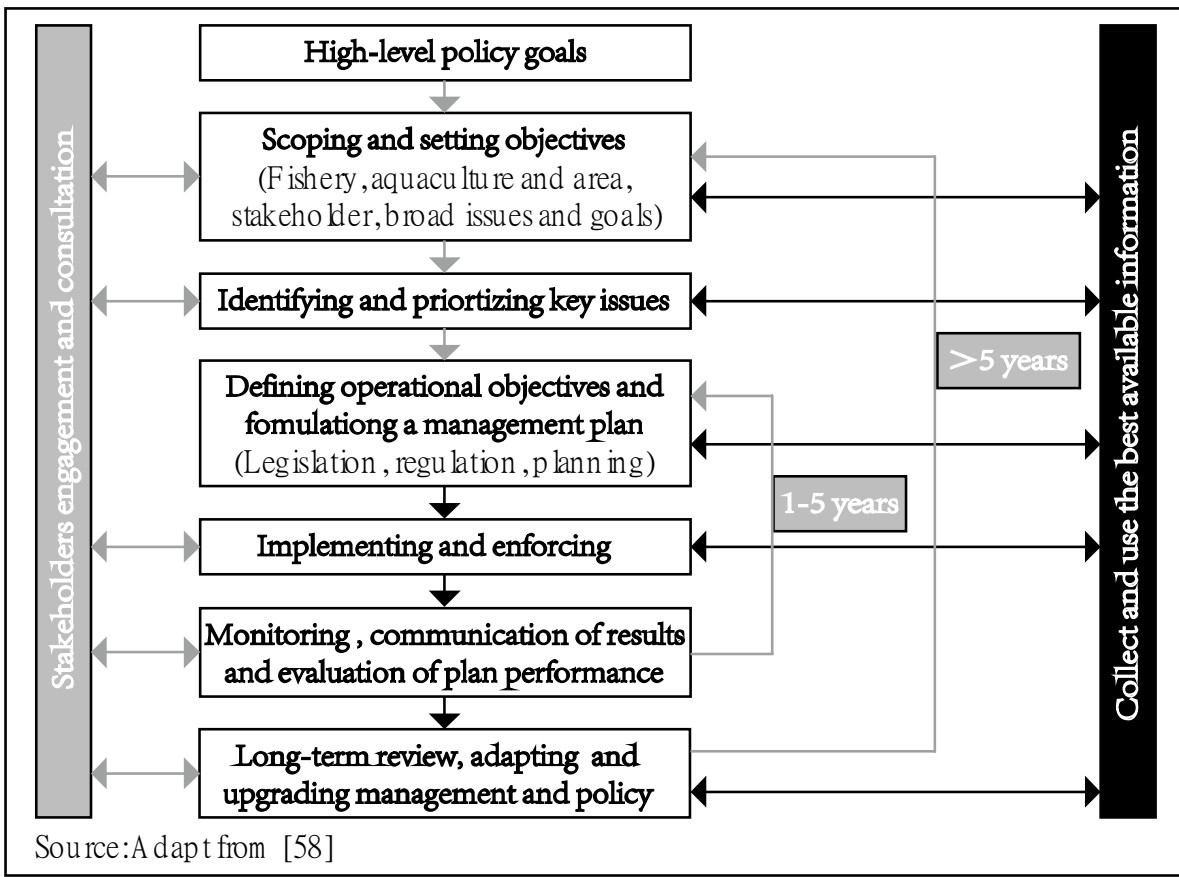

**Figure 6.** Planning process and implementation of the marine ranching management plan in the Ma'an Archipelago marine ranching pilot [58].

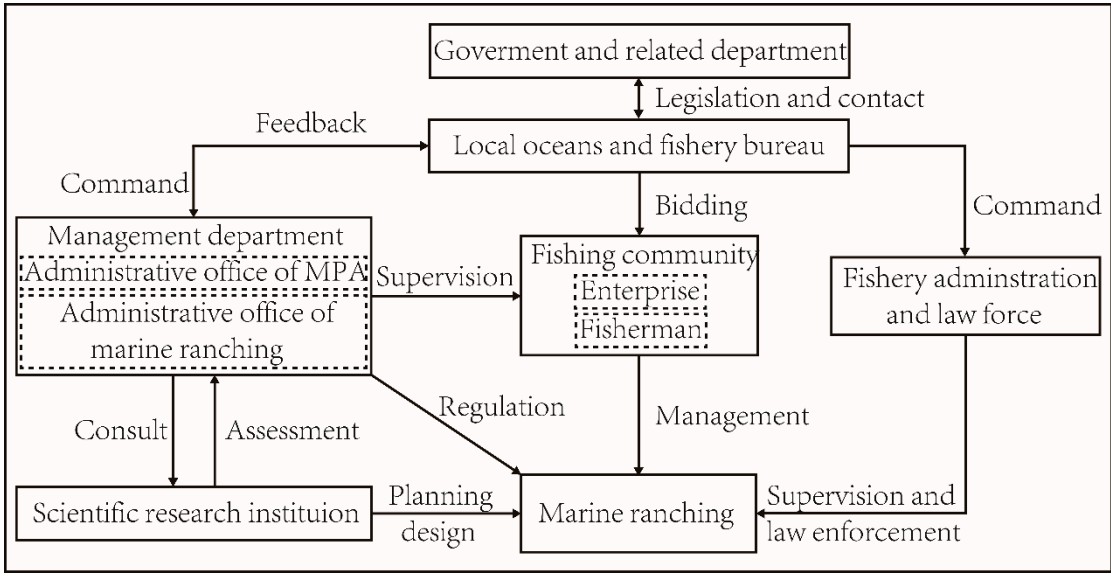

**Figure 7.** Regulatory system and relationships between stakeholders in the Ma'an marine ranching pilot.

**Table 2.** Management tools and measures for the different components of marine ranching.

| Management Tools and Measures |
| --- |
| **Input control** |
| Restriction or bans on certain fishing gear (including mesh size) or modes of fishing |
| Good stock enhancement practices, including artificial reefs and releasing |
| **Time/areas closures** |
| Zoning of areas of biological importance to wild caught seed or broodstock |
| Closed area (seasonal in typical habitats and permanently in MPA) |
| Protection and restoration of key seed settlements and nursery habitats |
| **Output control** |
| Restriction on certain protected or threatened species |
| **Market-related measures** |
| Traceability of products |
| Use of the Internet of Things and Internet + to consider of the potential power of consumer preferences |
| Good mariculture practices, including sustainable sourcing and use of feed |
| Certification system for geographical indications protection products |
| **Other measures** |
| Improve harvest, transport, and cultured practices to reduce wastage |
| Pollution controls |
| Control of disease |

## 4. Conclusions

In this review, the background of marine ranching, including history, high-level policy, and marine ranching construction activities, was analyzed, and four major techniques and representative cases of marine ranching construction in the ECS were reviewed. All concepts, theory, techniques, and application cases suggest that multiple fields can be integrated; the effects of ecological, economic, and social factors in marine ranching management can be quantitively simulated and assessed; and sustainable use can be achieved. Marine ranching is intuitively superior to the traditional fishery mode because it allows the inclusion of the advantages of EAA, EAF, and CBA, grows rapidly, and provides an increasing proportion of fishery products and ecosystem services for humans in China, in the ECS. However, because the available studies are still limited, it is not yet possible to implement tactical management in marine ranching in the ECS, and this will be an area for future progress in both conceptual exploration and applied ecology.

## 5. Synthesis and Future Directions

This review highlights the research on marine ranching in four continuous fields: (1) site selection and design provides marine spatial planning following ecological principles; (2) habitat restoration and construction technologies provide potential methods and procedures ranging from habitats to releasing; (3) stock enhancement and behavior control development, especially sound conditioning practices, are in the best interests of business and operators that acknowledge the concept of ranching; and (4) ecosystem-based models and real-time monitoring measures can be used for management decision-making, whereas the planning process, implementation, regulatory systems, and management tools can ensure the sustainable use of fishery resources, and considers the overall social and economic benefits [148]. Tackling the aforementioned fields will require building a strong foundation ranging from basic ecological process understanding to providing information for tactical decision-making, as well as single fishery and aquaculture yield, considering ecological–social–economic aspects [113]. Although several efforts and practices have been devoted to implementing marine ranching in the ECS, several challenges remain for additional research.

Although marine ranching construction in the ECS has considered ecosystem processes, many ecological processes may influence ecosystem functions in unknown ways. These efforts must now be complemented by studies that address two facets: (1) exploration of basic ecological processes, such as primary productivity, species interaction, material and energy flow between the different

trophic levels, species, seasonal and temporal dynamics (recruitment or movement), environment (hydrodynamic or environment factors) forcing, and anthropogenic disturbances (fishery or non-fishery activities) in the ecosystem; and (2) determining the inner relationships between ecological and genetic process, structures, and ecosystem functioning through partitioning ecological processes into statistical models, where care must be taken in their interpretation. The use of simple models for ecological processes and ecosystem considerations could become more widespread in the near future.

Another pressing but critical challenge, common to ecology in general, is whether and how insight from process-based model systems can be scaled up to complex marine ranching ecosystems [149]. So far, ecosystem models are not at the stage where a single model can be selected as a standard management or management procedure model that could be reliably applied at the tactical level to provide management recommendations in a particular case. In principle, three critical issues cause the considerable gap to exist between process models and the whole ecosystem model: (1) most studies on the ecological-process model usually focus on a single driving factor in the ecological process model through a closed experimental condition that can be intentionally isolated from dispersal, disturbances, and other relative processes to maximize experimental control [150]. However, most marine ranching ecosystems are not closed systems and the inputs and outputs of materials, matter, and the energy of the marine ranching ecosystem appear to be the norm in nature, whereas theory and experiments clearly predict that such exchange of materials can strongly modify the impact of diversity on ecosystem processes [151,152]. (2) Even if we clarified the input or output in the marine ranching ecosystem, the different parameters exhibit dynamic and compound interactions together as ecosystem complexity, whereas the best practice is to fit as much data (fixing and freeing parameters), so care must be taken in its interpretation [132]. Integrating more parameters into the ecosystem is an extremely difficult task as a result of the addition of interaction terms that are severely limited by the use of multi (or complex) factorial experiments, which is the foundation of the ecosystem model [153]. (3) The complexity of the ecosystem means that we cannot possibly expect to illustrate all the ecological processes we require in the marine ranching system, whereas a lack of full scientific certainty may cause serious irreversible damage to the ecosystem. As such, precautionary approaches are necessary to reduce some of the critical uncertainties of unknown ecological processes in the ecosystem models [154].

Except for unilateral ecological and environmental considerations, natural capital and ecosystem services assessment is another research hotspot. Understanding the form of ecosystem services, how to value them, the quantification and modeling of different categories of services to their monetary valuation, complex adaptive systems characterized by non-linear behaviors and changes, as well as identifying the limitations of such assessment methods, have become crucial questions that fully consider the interests of stakeholders and the public [155].

Given the emerging challenges and directions that we have outlined above, we are convinced that the management of marine ranching ecosystems will require scientists in different fields to embrace a broader suite of approaches than before. Several promising fields include: (1) aiming efforts toward ecological process explorations and conceptual models, which are important for identifying the relevant subsystems of marine ranching, the appropriate resolutions, and the essential processes for inclusion in tactical models and decision-making; (2) integrating dynamic-relative process-based models that include a range of ecosystem models (especially the variety of processes performed by different components in the food web), after management's strategic evaluation (MSE). Integrating the process-based model results into whole ecosystem models (such as Ecopath with Ecosim, GEEM (General Equilibrium Ecosystem Model), and ATANTIS) and dynamic system models (such as SEAPODYM (Spatial Ecosystem and Population Dynamics Model) and OSMOSE (Object-oriented Simulator of Marine ecOSystem Exploitation)). (3) Different methods could be used to solve parameter uncertainty, such as the Markov chain Monte Carlo (MCMC) to estimate the joint posterior probability of all model parameters conditioned on all the data, the expert judgement system for parameter selection, an explicit accounting of the numbers being estimated and fixed, the qualitative estimates of uncertainty in every parameter or sensitivity analysis for

quantifying parameter uncertainties, the strategic reduction of parameters by grouping parameters under a functional ecology concept, and fitting as much data as possible or time trends of data using appropriate likelihood structures in dynamic models. (4) Different techniques and alternative models could be used to explore the same decision-making issues and appropriate weight model plausibility, so that results from the most plausible models are given more weight in the decision-making process than results that are less plausible. (5) Bio-economic models such as Fisheries Library in R (FLR), methodological support for abio-economic model of population analysis of demersal resources (BIRDMOD), models for optimal sustainable effort in the seas (MOSES), and a fleet-based bio-economic simulation software for management strategies accounting for behavior of fisher (TEMAS) to assess the alternative management of fishery resources and welfare.

Ecosystem service assessment of marine ranching involves evaluating the values of ecosystem services in marine ranching, such as actual physical flows of ecosystem goods and services (e.g., yield and carbon sequestration), storm surge control, biodiversity conservation, ecological indicators, and other ecologically-oriented metrics based on thermodynamics and biophysical accounting to value ecosystems and their services, as well as the enhancement of recreation or tourism values of scenic beauty.

Several avenues exist for additional research but we will not attempt to cover them in detail in view of limited space; we will review them in the future. However, this does not mean that these areas are not important, for example: disease control, cultivation technology, mechanical harvest technology, integration of Internet + and the Internet of things to meet with consumer preferences, quality control of aquatic products and food securities, and efforts to combat climate change.

**Author Contributions:** Conceptualization, X.Z. (Xijie Zhou) and S.Z.; Methodology, X.Z. (Xijie Zhou) and X.Z. (Xu Zhao); Software, X.Z. (Xijie Zhou); Validation, X.Z. (Xu Zhao) and S.Z.; Formal analysis, X.Z. (Xijie Zhou) and X.Z. (Xu Zhao); Resources, X.Z. (Xijie Zhou); Data Curation, X.Z. (Xijie Zhou); Writing—Original Draft Preparation, X.Z. (Xijie Zhou) and S.Z.; Writing—Review & Editing, X.Z. (Xijie Zhou) and X.Z (Xu Zhao); Visualization, X.Z. (Xijie Zhou); Supervision, S.Z.; Project Administration, X.Z. (Xu Zhao); Funding Acquisition, S.Z. and J.L.

**Funding:** This research was supported by the National Natural Science Foundation of China (No. 41876191, 41176110, 41606146), Special Fund for Agro-scientific Research in the Public Interest (No. 201303047), and the China Agriculture Research System (CARS-50).

**Acknowledgments:** This research has been conducted with the financial support of the National Natural Science Foundation of China and China Agriculture Research System (CARS-50).

**Conflicts of Interest:** The authors declare no conflict of interest.

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
