# Peer review of "Marine Ranching Construction and Management in East China Sea: Programs for Sustainable Fishery and Aquaculture"

_water, doi:10.3390/w11061237_

Round 1

Reviewer 1 Report

This is a comprehensive review of conceptual frameworks, approaches and examples of marine ranching practices in the East China Sea. I found it interesting and informative and have mostly only minor comments for improvement. One overall suggestion is to increase the number of literature citations in the text as there are some sections where many facts are stated but without letting the reader where those facts come from. I mention some sections like this below, along with other general comments:

line 35: change to "traditional"

line 38: please change this line as it is a bit confusing.

line 45-46: I do not understand the meaning of this sentence.

line 50-57: I think the potential for sea ranching to include building of habitat structure should be included here. This is discussed later but it should also be included in the initial mention of definitions. The authors could consider using or adapting this definition which I have used in the past - "releasing juvenile specimens of species of fishery importance raised or reared in hatcheries and nurseries into the sea for subsequent harvest at the adult stage or manipulating fishery habitat to improve growth of the wild stocks" (Mustafa 2003, p. 142). Mustafa S (2003) Stock enhancement and sea ranching: objectives and potential. Rev Fish Biol Fish 13: 141−149

line 58-59: I would say that instead of categorising countries into eastern and western it would be better to just list the countries where examples come from, or use terms that are a bit more specific/formal. Later the terms "Asian" and "European" are used; maybe these could be used instead of "eastern" and "western".

line 63: what is meant by "the natural mode"?

around line 72: I am still trying to work out how exactly the Asian and European approaches differ. It would help if some specific examples or clearer description is given. e.g. it is stated that Asian countries use "artificial habitat based marine ranching", this is the same as the "artificial reef construction" that was explained as being done in western countries.

line 86-87: maybe change to "with the goals of reducing capture fishery yield".

line 126: please provide a reference for the statement here, and for the next sentence.

line 143: please provide references for all the figures stated in the above lines.

line 149-155: this long sentence is difficult to understand, please clarify it. Also, please explain better or reconsider including the term "ecological-function-outstanding".

line 188: delete "of", and change to "targets" in the next line.

Table one: I do not understand what is meant by "spatial based marine ranching". Is it meant to be "spatial planning based marine ranching"?

Table one: I think "extinct species" needs to be changed to "regionally extinct species".

Table one: in the last line of this table, should tools involving public or stakeholder engagement be included? These are often important to resolve conflicts.

line 209: change to "successfully".

line 240-242: I think this sentence could be toned down, e.g. "However, the evidence suggests that the use of this framework has been successful in several marine ranching settings, so it can be recommended that the steps proposed in Figure 2 may be implemented as proposed to meet the objectives of marine ranching, at least in the ECS."

lines 268-273: please list some references here.

Figure 4: please explain this figure a bit more, e.g. I do not understand what a "high artificial control-ability-link" means. What do the different levels of the triangles represent? What does it mean if they are different widths?

line 391: change to "restoration"

line 400-402: please cite a reference here describing the speed at which they are colonised.

line 418: I might be wrong but I do not think the term "bottom up cascade" works.

line 421: I do not understand what is meant by "the leakage".

line 428-444: please provide reference(s) for the facts in this paragraph.

line 470: please provide a proper reference here.

line 484: for me the word "cultural" is only associated with people, so here and elsewhere I would change "cultural" to "cultured".

line 550: add "the" in front of "same".

Fig. 5: in the first text of this figure change "date" to "data".

line 568: fix the error at the start of this line.

line 581: change "have" to "has"

line 591: I think "and numerical" can be deleted.

line 600: change to "its defect is that it is limited"

line 659: I think the word "planning" needs to be added between "spatial" and "following"

line 671: delete "ecosystem and"

line 677: change "determined" to "determining"

line 724: change "judge" to "judgment"

Author Response

Response to Reviewer 1 Comments

Dear Editors and Reviewer 1:

Thank you for your letter and reviewer’s comments concerning our manuscript entitled “Marine Ranching Construction and Management in East China Sea: Programs for Sustainable Fishery and Aquaculture” (ID: water-493706). Those comments are all valuable and very important for revising and improving our paper, as well as guiding significance to our researches. We have studied comments carefully and have made correction which we hope to meet with approval. Revised portions are marked in red in the paper. The main corrections in the paper and the response to the reviewer’s comments are as following:

Point 1: line 35: change to "traditional"

Response 1: We have made correction according to the reviewer’s comments. (line 35)

Point 2: line 38: please change this line as it is a bit confusing.

Response 2: We have used “Therefore, a new research area called the blue growth initiative (BGI) has attracted the attention in many fields of science” instead of “Therefore, a new research area called the blue growth initiative (BGI) incorporates technical expertise.” (line 38)

Point 3: line 45-46: I do not understand the meaning of this sentence.

Response 3: Our original target of this sentence is to connect the BGIs and the next sentence, which “The terms restocking, stock enhancement, culture-based fisheries, capture-based aquaculture, and sea ranching or marine ranching” can be considered as “given set of artificial activities and conditions”. However, since this sentence may cause confusion to readers and we have deleted the sentence. This change will not influence the content and framework (line 45-46)

Point 4: line 50-57: I think the potential for sea ranching to include building of habitat structure should be included here. This is discussed later but it should also be included in the initial mention of definitions. The authors could consider using or adapting this definition which I have used in the past - "releasing juvenile specimens of species of fishery importance raised or reared in hatcheries and nurseries into the sea for subsequent harvest at the adult stage or manipulating fishery habitat to improve growth of the wild stocks" (Mustafa 2003, p. 142). Mustafa S (2003) Stock enhancement and sea ranching: objectives and potential. Rev Fish Biol Fish 13: 141−149

Response 4: Thanks for your good advice, we have adapt the definition you used in the past “Mustafa defined sea ranching as “releasing juvenile specimens of species of fishery importance raised or reared in hatcheries and nurseries into the sea for subsequent harvest at the adult stage or manipulating fishery habitat to improve growth of the wild stocks [10]”, also ,we have adapt two more definition of marine ranching “Grant et al. suggested sea ranching consists of releasing hatchery-reared individuals into the wild, but with the expectation that individuals are harvested before they reproduce or mate with wild individuals [11]; While Kitada point out sea ranching aims at harvesting all juveniles (as possible) released in the harvesting areas, which can be managed by fishers[12];”(line 55-line 65)

Point 5: line 58-59: I would say that instead of categorising countries into eastern and western it would be better to just list the countries where examples come from, or use terms that are a bit more specific/formal. Later the terms "Asian" and "European" are used; maybe these could be used instead of "eastern" and "western".

Response 5: thanks for your suggestion, we have used “The major differences in the definition of marine ranching mainly exist between Asian countries and other countries (South America countries, European countries, and Oceanica countries” (line 69-line71)

Point 6: line 63: what is meant by "the natural mode"?

Response 6: This sentence may cause confusion to readers and we have deleted the sentence. To compare the difference between different areas, we have used some examples in the paper, see point 7 (line 78- line 79)

Point 7: around line 72: I am still trying to work out how exactly the Asian and European approaches differ. It would help if some specific examples or clearer description is given. e.g. it is stated that Asian countries use "artificial habitat based marine ranching", this is the same as the "artificial reef construction" that was explained as being done in western countries.

Response 7: We have used two compare examples to compare the definition and hot spot of marine ranching between Asian countries and European countries. The comparisons are as following: “For example, in South America countries, European and Oceanica countries, the expectation species for sea ranching are harvested before they reproduce or mate with wild individuals [7,11]. In contrast, Asian countries expected species for sea ranching return as adults to natal rearing areas to spawn [12].”(line 67-73) and “Furthermore, South America countries, Oceanica countries, and European countries refer to the use of stock enhancement and restocking strategies (i.e., releasing hatchery seeds to improve or rebuild fishery stocks) [8]. Most cases have shown that beneficiaries of ranching programs fall into four categories: releasing hatchery seeds to improve the self-sustaining populations, releasing hatchery seeds to rebuild severely depleted fish stocks, natural habitat conservation to maintain the habitat function of stock enhancement, and artificial reef construction to create an artificial hard substrate for reef fish [13]. In contrast except focused on developing effective hatchery, release, and field technologies for restocking and commercial wild restock yield management [9], East Asian countries have developed concentrated on a concept of aquaculture-based, artificial habitat-based marine ranching (e.g. buoyant raft and artificial reef) and rehabilitation-based marine ranching(e.g. seaweed bed, seagrass) based on advanced engineering, new materials and structures, which has become a hotspot worldwide since the late 2000s [12,14].(line 71-line 78)

Point 8: line 86-87: maybe change to "with the goals of reducing capture fishery yield".

Response 8: Thanks for your advice and we have changed the sentence to “with the goals of reducing capture fishery yield”. (line 105-line 106)

Point 9: line 126: please provide a reference for the statement here, and for the next sentence.

Response 9: we have added the reference “[24]” (now in line 147)

Point 10: line 149-155: this long sentence is difficult to understand, please clarify it. Also, please explain better or reconsider including the term "ecological-function-outstanding".

Response 10: We are sorry for the long sentence confused you, and we have cut the long sentence into short sentence “The Thirteenth Five-Year Plan declared that, for marine ranching, it would prioritize the regionally representative, high value of ecosystem service functions, scientific management, and social-economic benefits of modern marine ranching in one zone (coastal zone of China) and three regions (Yellow-Bohai Sea, East China Sea, and South China Sea. For marine ranching pilots construction, 178 national marine ranching demonstration areas will be constructed before 2020 (The total construction area is about 1000 km2 area, and 330 km2 of seagrass and seaweed bed will be restored or rehabilitated, and another 50 million stacked cubic meters artificial reef will be released).” (line 169- line 177)

Point 11: line 188: delete "of", and change to "targets" in the next line.

Response 11: Thanks for your advice. we have deleted "of", and change to "targets" in the next line. (line 209-line 210)

Point 12: Table one: I do not understand what is meant by "spatial based marine ranching". Is it meant to be "spatial planning based marine ranching"?

Response 12: Exactly, it meant to be "spatial planning based marine ranching" and we have used it instead of "spatial based marine ranching", thanks for your remind. (Table 1)

Point 13: Table one: I think "extinct species" needs to be changed to "regionally extinct species".

Response 13: Thanks for your advice, we have changed "extinct species" to "regionally extinct species". (Table 1)

Point 14: Table one: in the last line of this table, should tools involving public or stakeholder engagement be included? These are often important to resolve conflicts.

Response 14: Thanks for your suggestion, public and stakeholder engagement is very important and we have used it in “marine ranching management section”, for sure, it should be included here, and we have added it in (Table 1)

Point 15: line 209: change to "successfully".

Response 15: Thanks for your reminder and we have changed it to "successfully". (line 230)

Point 16: line 240-242: I think this sentence could be toned down, e.g. "However, the evidence suggests that the use of this framework has been successful in several marine ranching settings, so it can be recommended that the steps proposed in Figure 2 may be implemented as proposed to meet the objectives of marine ranching, at least in the ECS."

Response 16: Thanks for your good suggestion! We have used "However, the evidence suggests that the use of this framework has been successful in several marine ranching settings, so it can be recommended that the steps proposed in Figure 2 may be implemented as proposed to meet the objectives of marine ranching, at least in the ECS." (line 261-line 265) 

Point 17: lines 268-273: please list some references here.

Response 17: We have listed some reference in the paper, e.g. “[46-49]”. (line 304-line 309

Point 18: Figure 4: please explain this figure a bit more, e.g. I do not understand what a "high artificial control-ability-link" means. What do the different levels of the triangles represent? What does it mean if they are different widths?

Response 18: Thanks for your suggestion, we have revised the Figure 4.

Point 19: line 391: change to "restoration"

Response 19: Thanks for your reminder and we have changed it to “restoration”(line 448)

Point 20: line 400-402: please cite a reference here describing the speed at which they are colonised.

Response 20: we have added a reference “Chai et al. design a new artificial reef with pedestal and breeding broads and a procedure for macroalgae breeding were conducted in Gouqi Island, Zhejiang, China. The speed at which the artificial structures are colonized by plants and animals indicates that artificial habitats have the potential to extend ecosystem functions of natural habitats to offshore areas [83].”(line 458-line 462)

Point 21: line 418: I might be wrong but I do not think the term "bottom-up cascade" works.

Response 21: Thanks for your reminder, we have used “food-chain” instead of “bottom-up cascade”. (line 478)

Point 22: line 421: I do not understand what is meant by "the leakage".

Response 22: we have revised the sentence as “primary production diversity has strong positive bottom-up effects on secondary producer species (the leakage between primary producers and high trophic level organisms) that graze primary producers directly or extract organic matter for growth” (line 480-line 483)

Point 23: line 428-444: please provide reference(s) for the facts in this paragraph.

Response 23: We have added two references here [91] and [14], see line 493-line 507.

Point 24: line 470: please provide a proper reference here.

Response 24: We have provided three references here [83,105,106] line 539.

Point 25: line 484: for me the word "cultural" is only associated with people, so here and elsewhere I would change "cultural" to "cultured".

Response 25: Thanks for your good suggestion. We have used “cultured” here and in line 414, line 553, line 660 and Table 2

Point 26: line 550: add "the" in front of "same".

Response 26: Thanks for your reminder, we have added "the" in front of "same".(line 621)

Point 27: Fig. 5: in the first text of this figure change "date" to "data".

Response 27: Thanks for your reminder, we have changed "date" to "data". (Fig. 5)

Point 28: line 568: fix the error at the start of this line.

Response 28: We have fixed the error at the start of this line.

Point 29: line 581: change "have" to "has"

Response 29: Thanks for your good suggestion. We have changed “have” to “has”.(line 653)

Point 30: line 591: I think "and numerical" can be deleted.

Response 30: Thanks for your good suggestion. We have deleted “and numerical” in this sentence. (line 664)

Point 31: line 600: change to "its defect is that it is limited"

Response 31: Thanks for your good suggestion. We have changed “its defect is limited” to "its defect is that it is limited" (line 673).

Point 32: line 671: delete "ecosystem and"

Response 32: We have deleted “ecosystem and” (line 745)

Point 33: line 677: change "determined" to "determining"

Response 33: Thanks for your good suggestion. We have changed "determined" to "determining" (line 751)

Point 34: line 724: change "judge" to "judgment"

Response 34: Thanks for your good suggestion. We have changed "judge" to "judgment" (line 798)

Special thanks to you for your good comments.

Other changes:

We tried our best to improve the manuscript and made extensive editing of the English language and style in the manuscript. Meanwhile, we have adapted  information and supplement follow editor and reviewer 2’s suggestion as:  

Line 251: Fig 1. We have added exactly location of the study area.

Line 271-288 we have revised this paragraph as “Marine ranching construction can have ecosystem effects [35,36]. Take artificial reef construction as an example: artificial reefs are human-made structures installed in aquatic habitats that serve as a substrate and/or shelter for organisms, create exclusion areas to reduce the effort of industrial fishing and considered as an important method for ecosystem restoration [13,37,38]. Artificial reef construction may have both positive and negative environmental implications [39,40]. Although artificial reef construction can create suitable habitats for introduced (or target) species or attracting fishes, such as rocky fish, the shift in the substrate from a soft substrate to a hard substrate may negatively impact non-target species and habitats [41,42]. When site selection and design is poorly executed, artificial reef construction can affect ecosystem functions and services with negative environmental, social, and economic consequences [43]. Also, for the reef construction, artificial reefs deployed in close geographical proximity to other reefs can provide additional habitat to support complementary communities of fishes [44]; Becker suggests that close module spacing supports a connected assemblage and Yang suggest that the shape, material, configuration, and location of artificial reefs should be related with a specific goal to avoid mindless proliferation [45,46]. Therefore, marine ranching, as a sustainable fishery mode, demands construction suitability, sustainable practices at the level of the introduced (target) species, and taking responsibility for the interactions with local species (non-target species) in the ecosystem approaches context to minimize the negative consequence of blindly developing artificial reefs.”

Figure 3: We have revised Figure 3 by taking genetic effects into consideration.

Line 363-371:we have added some relative studies focus on genetic effects of cage aquaculture as “However, despite using ecosystem approaches, the debates of cage aquaculture mode still exist on following two aspects: (1) Cage aquaculture continues to be fed using large quantities of low-value (trash) fish from the wild, often comprising juveniles from potentially valuable species [57,58]; (2) some fish cultured in cages occasionally escape into the natural environment as a consequence of human error, net breakages, or natural causes such as predator or storm damage to cages[59,60]. Species like salmonid is likely to have considerable invasive and genetic impacts on local ecosystems [13,61-65]. To minimize the escape of cage aquaculture, acoustic conditioning can be used to recall escaped fish and when coupled with recapture, can reduce interactions with wild stocks and losses to the producer [66].”

Line 387-394: we have added some relative studies focus on genetic effects of stock enhancement as “Notably, compared to aquaculture mode, stock enhancement may have greater negative genetic effects on wild population, but the genetic effect diminished with the increased size of the wild population [70]. Given the complexity and uncontrollable risk of natural and artificial disturbances, there is still considerable uncertainty about the survival rate, recruitment, genetic effect, and quality during different life cycles of species release in the field and traditional enhancement and releasing cannot be successful over the long term unless sufficient efforts are also made to reduce harvest rates and rehabilitate natural habitats [70,71].”

Line 419-424: we have added some relative studies focus on genetic effects of IMTA as “ Meanwhile, although most species (e.g. scallop, abalone, sea cucumber, black sea bream et.al) of large scale IMTA programs in China show no genetic effect on adjacent area worldwide, however, hatchery producers in China will need to give consideration to future potential genetic impacts of juvenile releases where wild stocks are present, especially for those species (e.g. salmonid, red sea bream, drum, steelhead et al.) may have negative genetic effects on the ecosystem [12].

Line 435-438: we have added some relative studies focus on genetic effects of EAIMTCBA as “Also, as a systematic integration of habitat restoration, IMTA, and stock enhancement, a precautionary principle of both ecological and genetic effects is recommended particularly as concerns animal health checks, potential genetic and invasive effects prior to release into EAIMTCBA systems.”

Line 447-450: we have added the relative study to illustrate the importance of natural reserves, rehabilitation or restoration in marine ranching context as “The earlier view was dominated by the notion that natural reserves, rehabilitation or restoration were desirable because they would enhance the sustainability of habitats and the diversity of species. Kitada et al. suggest rehabilitation of the degraded natural habitats (e.g. nursery and spawning habitats) is the key to the success of marine ranching and stock enhancement [70].”

Line 483-486: we have added the relative study to illustrate the importance of natural reserves, rehabilitation or restoration of seaweed bed in marine ranching context as “(3) Seaweed communities should serve as nursery grounds due to abundant food source, and Kitada et al. suggest rehabilitation of the degraded nursery habitat is indispensable to recover depleted populations and minimize the negative genetic effects of stock enhancement [70].

Line 517-519: we have added the relative study to illustrate the importance of behavior control (or conditioning) of fishery resource in marine ranching context as “Efforts on behavior control (or conditioning) of fishery resource is another key to achieve sustainable fishery and aquaculture by minimizing the negative effects (both ecological and genetic) in ECS.”

Line 532-535: we have added the relative study to illustrate the importance of artificial reef structure in marine ranching context as “The fish behavior and fish use of reef structure can be applied in the allocation of artificial reef, such as selecting minimum buffering zone between reefs to more effectively enhance fisheries by minimizing attraction of fishes from existing reefs, while also maximizing food resource availability for reef fishes and area for routine reef fish behaviors[103].”

Line 572-573 we have added the relative study to illustrate the importance of comprehensive management in marine ranching context as “Rigorous monitoring of a large-scale marine stock enhancement program demonstrates the need for comprehensive management of fisheries [70]”

These changes will not influence the content and framework of the paper but improve our paper a lot. And here all the revises are using the "Track Changes" function in Microsoft Word. The number of lines and reference we mentioned is corresponding under the "Track Changes" function.

We appreciate for Editors and Reviewer’s warm work and hope that the correction will meet with approval.

Once again, thank you very much for your comments and suggestions.

Xijie Zhou, Xu Zhao, Shouyu Zhang, and Jun Lin

2019.05.21

Reviewer 2 Report

This paper reviews the history of marine ranching in China, construction techniques of marine ranching in the East China Sea (ECS), stock enhancement and behavioral control of fishery resources, marine ranching management. The focus was mainly on marine ranching construction techniques as shown in the article structure below:

1. Introduction

2. History of Marine Ranching Demonstration Regions Construction in China

3. Marine Ranching Construction in ECS: Techniques and Case

3.1. Marine Ranching Sites Selection and Design

3.2. Habitat Restoration and Construction Technologies

3.2.1. Overall Restoration and Construction Mode Use in the ECS

3.2.2. Restoration of Natural Habitat or Building Artificial Structure

3.2.3. Constructing Biotic Habitats Through Artificial Release of Biota of Different Trophic Levels in Artificial or Natural Habitats

3.3. Stock Enhancement and Behavioral Control of Fishery Resources

3.4. Marine Ranching Management

3.4.1. Marine Ranching Management Models

    3.4.2. Real-Time Monitoring and Long-Range Data Communication

 3.4.3. Management of Complex and Multi-Stakeholder Users and Uses

4. Conclusions

5. Synthesis and Future Direction

This paper was underpinned by the authors’ speculation that marine ranching is intuitively superior to the traditional fishery mode because marine ranching allows the inclusion of the advantages of the ecosystem approach to aquaculture (EAA), the ecosystem approach to fishery (EAF), and capture-based aquaculture (CBA), grows rapidly, and provides an increasing proportion of fishery products and ecosystem services for humans in China, in the ECS. The authors discussed on the future direction of marine ranching construction in the ECS, where uses of simulation models to evaluate marine ranching strategies were highlighted.

However, as mentioned by the author, empirical studies are limited, resulting this paper to be largely conceptual and it was unknown how EAA, EAF, and CBA were realized by marine ranching. In my view, marine ranching is a kind of put-and take fishery for released juveniles with artificial reefs providing newly created habitat. However, recent papers (e. g., Kitada 2018; Kitada et al. 2019) show that marine stock enhancement and sea ranching are not useful tools to sustain fisheries production in a long-term. Marine aquaculture also suffers from genetic effects of large-scale escapement from fish farms (e. g., Bolstad et al. 2017; Glover et al. 2017; Hagen et al. 2019). How do these results tolerate your conclusion? Simulation models may predict the effects of marine ranching, which however strongly depend on the model and can be adjusted by tuning the parameters with favorable outcomes. Readers of the manuscript may become anxious that the present enthusiasm of marine ranching misleads the future direction of sustainable fisheries in China.

An aim of reviews is to provide overview of the topic of interest, which can be achieved by compiling the empirical scientific studies such as an example of Yang et al. (2019). The definition of marine ranching was also poorly described without citing key literatures. Literatures of artificial reefs were lacking (e. g., Mills et al. 2017; Lee et al. 2018; Paxton et al. 2018; Rosemond et al. 2018; Becker et al. 2019; Cresson et al. 2019; Lima et al. 2019). Clavelle et al. (2019) also provided a synthesis of the interactions between mariculture and wild fisheries. Such literatures (or more) should be included in Introduction and the present status of marine ranching be reviewed.

Becker, A., Smith, J. A., Taylor, M. D., McLeod, J., & Lowry, M. B. (2019). Distribution of pelagic and epi-benthic fish around a multi-module artificial reef-field: Close module     spacing supports a connected assemblage. Fisheries Research, 209, 75-85.

Bolstad, G. H., Hindar, K., Robertsen, G., Jonsson, B., Sægrov, H., Diserud, O. H., ...  Barlaup, B. T. (2017). Gene flow from domesticated escapes alters the life history of wild Atlantic salmon. Nature Ecology and Evolution 1, 0124.

Clavelle, T., Lester, S. E., Gentry, R., & Froehlich, H. E. (2019). Interactions and management for the future of marine aquaculture and capture fisheries. Fish and Fisheries, 20(2), 368-388.

Cresson, P., Le Direach, L., Rouanet, E., Goberville, E., Astruch, P., Ourgaud, M., & Harmelin-Vivien, M. (2019). Functional traits unravel temporal changes in fish biomass production on artificial reefs. Marine Environmental Research.

Glover, K. A., Solberg, M. F., McGinnity, P., Hindar, K., Verspoor, E., Coulson, M. W., ... & Svåsand, T. (2017). Half a century of genetic interaction between farmed and wild Atlantic salmon: Status of knowledge and unanswered questions. Fish and Fisheries, 18(5), 890-927.

  Hagen, I. J., Jensen, A. J., Bolstad, G. H., Diserud, O. H., Hindar, K., Lo, H., & Karlsson,   S. (2019). Supplementary stocking selects for domesticated genotypes. Nature    

  Communications, 10 (1), 199.

Kitada, S. (2018). Economic, ecological and genetic impacts of marine stock enhancement and sea ranching: A systematic review. Fish and fisheries, 19(3), 511-532.

Kitada, S., Nakajima, K., Hamasaki, K., Shishidou, H., Waples, R. S., & Kishino, H. (2019). Rigorous monitoring of a large-scale marine stock enhancement program demonstrates the need for comprehensive management of fisheries and nursery habitat. Scientific reports, 9(1), 5290.

Lee, M. O., Otake, S., & Kim, J. K. (2018). Transition of artificial reefs (ARs) research and its prospects. Ocean & Coastal Management, 154, 55-65.

Lima, J. S., Zalmon, I. R., & Love, M. (2019). Overview and trends of ecological and socioeconomic research on artificial reefs. Marine Environmental Research.

Mills, K. A., Hamer, P. A., & Quinn, G. P. (2017). Artificial reefs create distinct fish assemblages. Marine Ecology Progress Series, 585, 155-173.

Paxton, A. B., Revels, L. W., Rosemond, R. C., Van Hoeck, R. V., Lemoine, H. R., Taylor, J. C., & Peterson, C. H. (2018). Convergence of fish community structure between a newly deployed and an established artificial reef along a five-month trajectory. Ecological Engineering, 123, 185-192.

Rosemond, R. C., Paxton, A. B., Lemoine, H. R., Fegley, S. R., & Peterson, C. H. (2018). Fish use of reef structures and adjacent sand flats: implications for selecting minimum buffer zones between new artificial reefs and existing reefs. Marine Ecology Progress Series, 587, 187-199.

Yang, X., Lin, C., Song, X., Xu, M., & Yang, H. (2019). Effects of artificial reefs on the meiofaunal community and benthic environment-A case study in Bohai Sea, China. Marine pollution bulletin, 140, 179-187.

Author Response

Response to Reviewer 2 Comments

Dear Editors and Reviewer 2:

Thank you for your letter and reviewer’s comments concerning our manuscript entitled “Marine Ranching Construction and Management in East China Sea: Programs for Sustainable Fishery and Aquaculture” (ID: water-493706). Those comments are all valuable and very important for revising and improving our paper, as well as guiding significance to our researches. We have studied comments carefully and have made correction which we hope to meet with approval. Revised portions are marked in red in the paper. The main corrections in the paper and the response to the reviewer’s comments are as following:

Point 1: However, as mentioned by the author, empirical studies are limited, resulting this paper to be largely conceptual and it was unknown how EAA, EAF, and CBA were realized by marine ranching.

Response 1: Thank you for reviewer 2's comments. As reviewer 2 mentioned, we have stated that empirical studies are limited, however, if reviewer 2 has misunderstood our conclusion: “However, because the available studies are still limited, it is not yet possible to implement tactical management in marine ranching in the ECS (line 728-730)” in the Conclusion section. This conclusion comes from the “In China, although tactical models are rarely reported for making tactical decisions, some management practices in the ECS that been implemented as a continuum running from conceptual understanding and long-term strategic decision-making to tactical decision-making… (line 638-641)” Honestly, tactical management examples are still limited in the ECS (line 639-661). However, except studies on tactical management, most of the empirical studies focus on conceptual exploring and strategic decision-making in ECS, for your convenience, you may find the highlighted parts in the following sections (e.g. Marine Ranching Sites Selection and Design, empirical studies cases: line 229-249; Habitat Restoration and Construction Technologies, overall construction situation: line319-331, empirical studies on cage aquaculture: line 355-371; empirical studies on IMTA: line 397-415; empirical studies on Restoration of Natural Habitat or Building Artificial Structure: line 444-462; empirical studies on Artificial Release of Biota of Different Trophic Levels in Artificial or Natural Habitats :line 468-507; empirical studies on artificial substrate and reef structures and fishery resources line526-550; empirical studies on acoustic conditioning line 556-565; empirical studies on Marine Ranching Management Models line 588-605 and line 610-623…),

Amongst above empirical studies focus on conceptual exploring and strategic decision-making, marine spatial planning, stock enhancement, cage aquaculture, IMTA, habitat restoration and rehabilitation, acoustic conditioning, conceptual, strategic and tactical models, and management of complex and multi-stakeholder users and uses are important tools for EAA, EAF, and CBA(according to several FAO’s annual reports and technical guidelines [2-4], [30], [54], [57,58], [74], [116], [125]) which have solved how EAA, EAF, and CBA were realized by marine ranching. We believe the relative empirical studies we mentioned above can support how marine ranching realize EAA, EAF, and CBA in ECS to some degree.

Point 2: In my view, marine ranching is a kind of put-and take fishery for released juveniles with artificial reefs providing newly created habitat. However, recent papers (e. g., Kitada 2018; Kitada et al. 2019) show that marine stock enhancement and sea ranching are not useful tools to sustain fisheries production in a long-term. How do these results tolerate your conclusion?

Response 2: Thank you for the advice and recommendation for references. We have learned more about marine ranching and rewritten. and we have taken full consideration on the researches of Kitada 2018 and Kitada et al. 2019 in our paper. We have learned more about marine ranching and rewritten corresponding sections, added several pieces of literature of marine ranching definition and comparison of marine ranching between different countries were conducted.

For the definition of marine ranching, after reviewed many studies on marine ranching and sea ranching, we found there is still no uniform definition on “sea ranching”. You may find our statement about the main reason “as there are huge differences in the state of the marine environment, fish stock, fishery products demand, technology level, and marine policy between different countries and regions, and as marine ranching is an important activity, marine ranching (or sea ranching or ocean ranching) is defined heterogeneously in the line 65-68.

The definition shared by reviewer 2 is similar with one of many scientists in European countries, South American countries and Oceanic countries as “marine ranching was defined as the release of cultured juveniles into marine and estuarine environments for subsequent growth and harvest, with no intention that the released animals should contribute to the spawning biomass (though this can occur) (line 56-58)” and “Grant et al. suggested sea ranching consists of releasing hatchery-reared individuals into the wild, but with the expectation that individuals are harvested before they reproduce or mate with wild individuals[11]”. In these countries, the expectation species for sea ranching are harvested before they reproduce or mate with wild individuals.

However, in Asian countries, “Mustafa defined sea ranching as “releasing juvenile specimens of species of fishery importance raised or reared in hatcheries and nurseries into the sea for subsequent harvest at the adult stage or manipulating fishery habitat to improve growth of the wild stocks[10] (line 58-61)” while “Kitada point out sea ranching aims at harvesting all juveniles (as possible) released in the harvesting areas, which can be managed by fishers [12] (line 63-65)”. Also, “except focused on developing effective hatchery, release, and field technologies for restocking and commercial wild restock yield management [9], East Asian countries have developed concentrated on a concept of aquaculture-based, or artificial habitat-based marine ranching(e.g. buoyant raft and artificial reef) and rehabilitation-based marine ranching (e.g. seaweed bed and seagrass) based on advanced engineering, new materials and structure[12,-14] (line 85-90)”. Our paper is underpinned by the most accepted definition of marine ranching in China and we cited the most widely accepted definition of marine ranching in China(e.g., a sustainable fishery mode that facilitates the breeding and conservation of fishery resource and marine eco-environment improvement using various measures such as artificial reef, stock enhancement, and releasing, to construct or restore breeding, growth, forage, and shelter habitat for marine organisms with ecosystem perspectives and principles in certain sea areas [17] (line111 -line114).

With reviewer 2's kind reminder, we have learned the Kitada et al.'s paper (2019) carefully and added it in our paper. However, we found the following precondition before his conclusion “Our results represent crucial evidence that hatcheries for enhancement and conservation of populations cannot be successful over the long term unless sufficient efforts are also made to reduce harvest rates and rehabilitate natural habitats. (in Kitada et al. 2019 Abstract section). We wonder if reviewer 2 has missed this precondition “unless sufficient efforts are also made to reduce harvest rates and rehabilitate natural habitats”. In other words, Kitada believes that if efforts can be developed in the comprehensive management and rehabilitate natural habitats,  sustainable fishery can be achieved.

Meanwhile, we kindly ask reviewer 2 check the following parts again which we have mentioned in the main text: In section 2. History of Marine Ranching Demonstration Regions Construction in China, “The Thirteenth Five-Year Plan declared that…and 330 km2 of seagrass and seaweed bed will be restored or rehabilitated… (line 169-175)”; in section 3.2. Habitat Restoration and Construction Technologies, we have highlighted and reviewed many empirical studies and efforts to improve rehabilitate natural habitats in ECS and efforts devoted in East China Sea as “This section reviews the various techniques and issues to be considered in relation to habitat restoration and construction, inclusive of natural habitat (seaweed beds, mangroves, seagrass) conservation or restoration and artificial habitats construction (e.g., artificial reefs, cages, buoyant rafts, artificial seaweed, mangroves, and seagrass habitats) [46-49] (line 304-309)…”; in section 3.2.2 Restoration of Natural Habitat or Building Artificial structure, we provide empirical studies of restoration of natural habitat or building artificial structures as “To stop the degradation trend, new natural or artificial habitats can intentionally or unintentionally be created [77]. The earlier view was dominated by the notion that natural reserves, rehabilitation or restoration were desirable because they would enhance the sustainability of habitats and the diversity of species…To restore the natural habitat…leading to numerous attempts to actively add artificial habitats into the ecosystem while restoration of degraded reefs occurs [81]. In contrast, a popular view is that artificial habitats are part of the toolkit for managing the allocation and sustainability of harvestable resources [82]… Chai et al. design a new artificial reef with pedestal and breeding broads and a procedure for macroalgae breeding were conducted in Gouqi Island, Zhejiang, China [83](line 440-462)” and “Macroalgae are the marine foundation species that: (1) modify the physical environment by creating canopy- or turf-forming forests on natural habitats (rocky reefs) or artificial culture structure… and (2) provides material resources (400 times net primary productivity (NPP) than phytoplankton), and food sources for both the grazing chain and detritus chain [86]…Namely, primary production diversity has strong positive bottom-up effects on secondary producer species (the leakage between primary producers and high trophic level organisms) that graze primary producers directly or extract organic matter for growth [88]; (3) Seaweed communities should serve as nursery grounds due to abundant food source [70] (line 465-490) ”.

And as for comprehensive management aspects, the reviewer may miss some information as in section IntroductionThe government of China, along with the FAO, have proposed the Thirteenth Five-Year Plan for Economic and Social Development of the People’s Republic of China (2016–2020), with the goals of reducing capture fishery yieldsubstantially reducing the growth rate of aquaculture production, and developing marine ranching as a major mode of regional comprehensive and sustainable development....(line 102-110) ” and “Marine ranching management frameworks provide technologies to place fisheries and aquaculture activities within the broader ecosystem context, ensuring that all stakeholders take full part in decision-making and in the implementation of appropriate measures and regulations…(line 569-570)”; In section 3.4.1. Marine Ranching Management Models, “Protection and restoration of key seed settlements and nursery habitats” is noticed as an important part in marine ranching (Table 2, line 718).

We really appreciate for the kind reminder of such important literature (Kitada 2018, Kitada et al. 2019) should be included in our paper, we have added their views and conclusions in Introduction sectionKitada point out sea ranching aims at harvesting all juveniles (as possible) released in the harvesting areas, which can be managed by fishers [12] (line 63-65)”; in section 3.2.2 Restoration of Natural Habitat or Building Artificial Structure (line 449-450) “Kitada et al. suggest rehabilitation of the degraded natural habitats (e.g. nursery and spawning habitats) is the key to success of marine ranching and stock enhancement [70]”, ” (3) Seaweed communities should serve as nursery grounds due to abundant food source, and Kitada et al. suggest rehabilitation of the degraded nursery habitat is indispensable to recover depleted populations and minimize the negative genetic effects of stock enhancement [70] (line 483-486)” and “Rigorous monitoring of a large-scale marine stock enhancement program demonstrates the need for comprehensive management of fisheries [70] (line 572-573)”and we believe Kitada’s results could tolerate our conclusion.

For your convenience, the newly added references have been highlighted and listed as following.

[10] Mustafa, S. Stock enhancement and sea ranching: objectives and potential. Rev Fish Biol Fish.2003, 13: 141−149

[11] Grant, W.S., Jasper, J., Bekkevold, D., Adkison, M. Responsible genetic approach to stock restoration, sea ranching and stock enhancement of marine fishes and invertebrates. Rev Fish Biol Fish. 2017, 27, 615–649.

[12] Kitada, S. Economic, ecological and genetic impacts of marine stock enhancement and sea ranching: A systematic review. Fish and Fisheries. 2018, 19,511–532.

[13] Lima, J.S., Zalmon, I.R., Love, M. Overview and trends of ecological and socioeconomic research on artificial reefs. Marine Environmental Research 2019, 145,81-96.

[46] Yang, X.Y., Lin, C.G., Song, X.Y., Xu, M., Yang, H.S. Effects of artificial reefs on the meiofaunal community and benthic environment - A case study in Bohai Sea, China. Marine Pollution Bulletin.2019, 140, 179-187.

[47] Peng, Y.S., Zheng, M.X., Zheng, Z.X., Wu, G.C., Chen, Y.C., Xu, H.L., Tian, G.H., Peng, S.H., Chen, G.Z. Lee, S.Y. Virtual increase or latent loss? A reassessment of mangrove populations and their conservation in Guangdong, southern China. Marine Pollution Bulletin.2016, 109, 691-699.

[48] Zhang, X.M., Zhou, Y., Liu, P., Wang, F., Liu, B.J., Liu, X.J., Yang, H.S. Temporal pattern in biometrics and nutrient stoichiometry of the intertidal seagrass Zostera japonica and its adaptation to air exposure in a temperate marine lagoon (China): Implications for restoration and management. Marine Pollution Bulletin.2015, 94, 103-113.

[49] Liu, Z.Z., Cui, B.S., He, Q. Shifting paradigms in coastal restoration: Six decades' lessons from China. Science of the Total Environment. 2016, 566-567.

[70] Kitada, S., Nakajima, K., Hamasaki, K., Shishidou, H., Waples, R. S., & Kishino, H. Rigorous monitoring of a large-scale marine stock enhancement program demonstrates the need for comprehensive management of fisheries and nursery habitat. Scientific reports.2019, 9, 5290.

Point 3: Marine aquaculture also suffers from genetic effects of large-scale escapement from fish farms (e. g., Bolstad et al. 2017; Glover et al. 2017; Hagen et al. 2019). How do these results tolerate your conclusion?

Response 3: We appreciate your good advice and we believe genetic effects of aquaculture (e.g. cage, IMTA), stock enhancement activities are also an important part in marine ranching. Although we have mentioned relevant research in ECS in reported by Cheng and Jiang (2010), they suggested “(2) fully considering ecosystem perspectives, including ensuring that wild population adaption and genetic diversity will not be degraded and decreased after release of the artificial breed (in section 3.2.1. Overall Restoration and Construction Mode Use in the ECS, line -line)”, however, that is far from enough for the genetic effects problem. Hence, based on  reviewer 2’s constructive comments, we have added and revised several contexts in this paper to tackle the genetic issues:

For genetic effects of cage culture, we have added supplement as “(2) some fish cultured in cages occasionally escape into the natural environment as a consequence of human error, net breakages, or natural causes such as predator or storm damage to cages [59,60]. Species like salmonid is likely to have considerable invasive and genetic impacts on local ecosystems [13,61-65]. To minimize the escape of cage aquaculture, acoustic conditioning can be used to recall escaped fish and when coupled with recapture, can reduce interactions with wild stocks and losses to the producer [66] line 366-371”.

For genetic effects of stock enhancement, we have added supplements as “Notably, compared to aquaculture mode, stock enhancement may have greater negative genetic effects on wild population, but the genetic effect diminished with the increased size of the wild population [70]. Given the complexity and uncontrollable risk of natural and artificial disturbances, there is still considerable uncertainty about the survival rate, recruitment, genetic effect, and quality during different life cycles of species release in the field and traditional enhancement and releasing cannot be successful over the long term unless sufficient efforts are also made to reduce harvest rates and rehabilitate natural habitats [70,71]. line 387-394”. Meanwhile study aims to minimize the genetic effects of stock enhancement have been added as follow: “(3) Seaweed communities should serve as nursery grounds due to abundant food source, and Kitada et al. suggest rehabilitation of the degraded nursery habitat is indispensable to recover depleted populations and minimize the negative genetic effects of stock enhancement [70] line 483-486” And “Efforts on behavior control (or conditioning) of fishery resource is another key to achieve sustainable fishery and aquaculture by minimize the negative effects (both ecological and genetic) in ECS… line 517-565.

For genetic effects of IMTA and marine ranching, we have added supplement as “Meanwhile, although most species (e.g. scallop, abalone, sea cucumber, black sea bream et.al) of large scale IMTA programs in China show no genetic effect on adjacent area, however, hatchery producers in China will need to give consideration to future potential genetic impacts of juvenile releases where wild stocks are present, especially for those species (e.g. salmonid, red sea bream, drum, steelhead et al.) may have negative genetic effects on the ecosystem [12]. Line419 -line 424” and “Also, as a systematic integration of habitat restoration, IMTA, and stock enhancement, a precautionary principle of both ecological and genetic effects is recommended particularly as concerns animal health checks, potential genetic and invasive effects prior to release into EAIMTCBA systems [13] line 435-438”.

We also revised  Figure 3 by taking genetic effects into consideration

Thanks a lot for reviewer 2’s constructive suggestions on genetic effects, they are very helpful for us to improve our paper.

For your convenience, the newly added references have been highlighted and listed as following.

[13] Lima, J.S., Zalmon, I.R., Love, M. Overview and trends of ecological and socioeconomic research on artificial reefs. Marine Environmental Research 2019, 145,81-96.

[59] Dempster, T., Arechavala-Lopez, P., Barrett, L.T., Fleming, L.A., Sanchez-Jerez, P., Uglem, I. Recapturing escaped fish from marine aquaculture is largely unsuccessful: alternatives to reduce the number of escapees in the wild. Reviews in Aquaculture.2018, 10,153-167.

[60] Skilbrei, O.T., Wennevik, V. The use of catch statistics to monitor the abundance of escaped farmed Atlantic salmon and rainbow trout in the sea. ICES J. Mar. Sci. 2006, 63, 1190–1200.

[61] Abrantes, K.G., Lyle, J.M., Nichols, P.D., Semmens, J.M. Do exotic salmonids feed on native fauna after escaping from aquaculture cages in Tasmania, Australia? Canadian Journal of Fisheries & Aquatic Sciences.2011, 68, 1539-1551.

[62] Bolstad, G. H., Hindar, K., Robertsen , G., Jonsson, B., Sægrov, H., Diserud, O. H., Fishe, P., Jensen, A.J., Naesje, T.F., Barlaup, B.T., Floro-Larsen, B., Lo, H., Niemela, E., Karlsson, S. Gene flow from domesticated escapes alters the life history of wild Atlantic salmon. Nature Ecology and Evolution.2017, 1, 0124..

[63] Glover, K. A., Solberg, M. F., McGinnity, P., Hindar, K., Verspoor, E., Coulson, M. W., Hansen M.M., Araki, H., Skaala, Y. Svåsand, T. Half a century of genetic interaction between farmed and wild Atlantic salmon: Status of knowledge and unanswered questions. Fish and Fisheries.2017, 18, 890-927.

[64] Hagen, I. J., Jensen, A. J., Bolstad, G. H., Diserud, O. H., Hindar, K., Lo, H., & Karlsson, S. Supplementary stocking selects for domesticated genotypes. Nature Communications. 2019,10, 1-8.

[65] Clavelle, T., Lester, S.E., Gentry, R., Froehlich, H.E. Interactions and management for the future of marine aquaculture and capture fisheries. Fish Fish. 2019, 20,368–388.

[66] Tlusty, M.F., Andrew, J., Baldwin, K., Bradley, T.M. Acoustic conditioning for recall/recapture of escaped Atlantic salmon and rainbow trout. Aquaculture.2008, 274,57-64.

[70] Kitada, S., Nakajima, K., Hamasaki, K., Shishidou, H., Waples, R. S., & Kishino, H. Rigorous monitoring of a large-scale marine stock enhancement program demonstrates the need for comprehensive management of fisheries and nursery habitat. Scientific reports.2019, 9, 5290.

Point 4: Simulation models may predict the effects of marine ranching, which however strongly depend on the model and can be adjusted by tuning the parameters with favorable outcomes.

Response 4: Thank you for your comments. We fully understand your anxious about ethics of model researches and we heavily opposed with this kind of fraud.Indeed, models can be adjusted by tunning the parameters with favourable outcomes, but it is forbidden. To solve this kind of scientific fraud, we have discussed as following in the original manuscript: (1)Sometimes different models can be used to answer the same strategy problem, and models providing strategic advice can be selected according to the data basis, where a strategic model selection example was established by Zhang et al. (Figure 5) [14].”; (2) Different methods could be used to solve parameter uncertainty, such as the Markov Chain Monte Carlo (MCMC) to estimate the joint posterior probability of all model parameters conditioned on all the data, the expert judgement system for parameter selection, an explicit accounting of the numbers being estimated and fixed, the qualitative estimates of uncertainty in every parameter or sensitivity analysis for quantifying parameter uncertainties, the strategic reduction of parameters by grouping parameters under a functional ecology concept, and fitting as much data as possible or time trends of data using appropriate likelihood structures in dynamic models. (line796-line803)”; and (3) “(4) Different techniques and alternative models could be used to explore the same decision-making issues and appropriate weight model plausibility, so that results from the most plausible models are given more weight in the decision-making process than results that are less plausible. (line 803-line 806)”.

Point 5: An aim of reviews is to provide overview of the topic of interest, which can be achieved by compiling the empirical scientific studies such as an example of Yang et al. (2019). The definition of marine ranching was also poorly described without citing key literatures. Literatures of artificial reefs were lacking (e. g., Mills et al. 2017; Lee et al. 2018; Paxton et al. 2018; Rosemond et al. 2018; Becker et al. 2019; Cresson et al. 2019; Lima et al. 2019). Clavelle et al. (2019) also provided a synthesis of the interactions between mariculture and wild fisheries. Such literatures (or more) should be included in Introduction and the present status of marine ranching be reviewed.

Response 5: Thank you for your valuable advice and recommendation on latest important researches about marine ranching and artificial reef, which have helped a lot to improve quality of the paper.

We have added relative studies in our paper. Specifically, firstly, we have added and compiled several empirical scientific studies such as yang et al. (2019) [46], Peng et al. (2016) [47], Zhang et al. (2015) [48], Liu et al. (2015) [49], Chai et al. (2014) [83], Wang et al. (2016) [91], Huang et al. (2016) [105], Liu et al. (2013) [106]. Secondly, we have mentioned and cited many empirical studies in ECS, see the response to Point 1

Meanwhile, we have rewritten the context of artificial reefs in section 3.1. Marine Ranching Sites Selection and Design as “Marine ranching construction can have ecosystem effects [35,36]. Take artificial reef construction as an example: artificial reefs are human-made structures installed in aquatic habitats that serve as a substrate and/or shelter for organisms, create exclusion areas to reduce the effort of industrial fishing and considered as an important method for ecosystem restoration [13,37,38]. Artificial reef construction may have both positive and negative environmental implications [39,40]. Although artificial reef construction can create suitable habitats for introduced (or target) species or attracting fishes, such as rocky fish, the shift in substrate from a soft substrate to a hard substrate may negatively impact non-target species and habitats [41,42]. When site selection and design is poorly executed, artificial reef construction can affect ecosystem functions and services with negative environmental, social, and economic consequences [43]. Also, for the reef construction, artificial reefs deployed in close geographical proximity to other reefs can provide additional habitat to support complementary communities of fishes [44]; Becker suggests that close module spacing supports a connected assemblage and Yang suggest that the shape, material, configuration and location of artificial reefs should be related with a specific goal to avoid mindless proliferation [45,46].Therefore, marine ranching, as a sustainable fishery mode, demands construction suitability, sustainable practices at the level of the introduced (target) species, and taking responsibility for the interactions with local species (non-target species) in the ecosystem approaches context to minimize the negative consequence of blindly developing artificial reefsline 271-288” And “As a result, artificial reefs and cultural structures can provide abundant food sources, suitable nursery structures to hide from predators, and aggregate fishery resources due to the behavioral preferences of species in many sea areas. Reef fish, such as black seabream (Sparus macrocephalus) and red seabream (Chrysophrys major) in the ECS, are two typical examples that show that artificial structures can coincide with the forage and hiding behaviors of certain weak migratory species, respectively [98,99]. Several studies from around the world, such as studies on reef fishes associated with artificial reefs in the northwest Gulf of Mexico, gastropod in seaweed beds, and grunt fish (Haemulon Sciurus and Haemulon flavolineatum) in the backreef habitat, seem to be reaching similar conclusions [100-102]. The fish behavior and fish use of reef structure can be applied in the allocation of artificial reef, such as selecting minimum buffering zone between reefs to more effectively enhance fisheries by minimizing attraction of fishes from existing reefs, while also maximizing food resource availability for reef fishes and area for routine reef fish behaviors [103] line 523-535”.

What's more, we have compiled Clavelle et al. (2019)’s research in section 3.2.1. Overall Restoration and Construction Mode Use in the ECS (“(2) some fish cultured in cages occasionally escape into the natural environment as a consequence of human error, net breakages, or natural causes such as predator or storm damage to cages [59,60]. Species like salmonid is likely to have a considerable invasive and genetic impacts on local ecosystems [13,61-65]. To minimize the escape of cage aquaculture, acoustic conditioning can be used to recall escaped fish and when coupled with recapture, can reduce interactions with wild stocks and losses to the producer [66] line 366-371

We appreciate for your good suggestion and rewritten or added many key literatures in this paper, and we have listed them under this paragraph, thanks a lot.

For your convenience, hereafter are the literatures we added in the paper (additional 33 reference):

[10] Mustafa, S. Stock enhancement and sea ranching: objectives and potential. Rev Fish Biol Fish.2003, 13: 141−149

[11] Grant, W.S., Jasper, J., Bekkevold, D., Adkison, M. Responsible genetic approach to stock restoration, sea ranching and stock enhancement of marine fishes and invertebrates. Rev Fish Biol Fish. 2017, 27, 615–649.

[12] Kitada, S. Economic, ecological and genetic impacts of marine stock enhancement and sea ranching: A systematic review. Fish and Fisheries. 2018, 19,511–532.

[13] Lima, J.S., Zalmon, I.R., Love, M. Overview and trends of ecological and socioeconomic research on artificial reefs. Marine Environmental Research 2019, 145,81-96.

[24] Si, Y.J. Legal regulation of artificial reef. Ocean University of China, 2012(in Chinese)

[26] Fisheries Administration Bureau of Ministry of Agriculture and Rural Affairs, Chinese Academy of Fishery Sciences. Research on the Development Strategy of Marine Ranching in China. Beijing: China Agriculture Press.2017(in Chinese)

[36] Bulleri, F., Chapman, M.G. The introduction of coastal infrastructure as a driver of change in marine environments. Appl. Ecol.2010, 47, 26–35.

[37] Firth, L.B., Knights, A.M., Bridger, D., Evans, A.J., Mieszkowska, N., Moore, P.J., O'Connor, N.E., Sheehan, E.V., Thompson, R.C., Hawkins, S.J. Ocean sprawl: challenges and opportunities for biodiversity management in a changing world. Oceanogr. Mar. Biol. Annu. Rev. 2016, 54, 193–269.

[38] Grossman, G.D., Jones, G.P., Seaman, W.J. Do artificial reefs increase regional fish do artificial reefs increase regional fish Production? A review of existing data. Fisheries.1997, 22, 17–23.

[39] Smith, J.A., Lowry, M.B., Suthers, I.M. Fish attraction to artificial reefs not always harmful: a simulation study. Ecol. Evol. 2015,5, 4590–4602.

[40] Mills, K. A., Hamer, P. A., & Quinn, G. P. Artificial reefs create distinct fish assemblages. Marine Ecology Progress Series.2017, 585, 155-173.

[41] Lee, M. O., Otake, S., & Kim, J. K., Transition of artificial reefs (ARs) research and its prospects. Ocean & Coastal Management 2018, 154, 55-65

[42] Brickhill, M.J., Lee, S.Y., Connolly, R.M. Fishes associated with artificial reefs: attributing changes to attraction or production using novel approaches. J. Fish. Biol.2005, 67, 53–71.

[44] Aguilera, M.A., Broitman, B.R., Thiel, M. Artificial breakwaters as garbage bins: structural complexity enhances anthropogenic litter accumulation in marine intertidal habitats. Environ. Pollut. 2016,214, 737–747.

[45] Paxton, A.B., Revels, L.W., Rosemond R.C., Hoeck, R.V., Lemoine, H.R., Taylor, J.C., Peterson, C.H. Convergence of fish community structure between a newly deployed and an established artificial reef along a five-month trajectory. Ecological Engineering.2018, 123, 185-192.

[46] Yang, X.Y., Lin, C.G., Song, X.Y., Xu, M., Yang, H.S. Effects of artificial reefs on the meiofaunal community and benthic environment - A case study in Bohai Sea, China. Marine Pollution Bulletin.2019, 140, 179-187.

[47] Peng, Y.S., Zheng, M.X., Zheng, Z.X., Wu, G.C., Chen, Y.C., Xu, H.L., Tian, G.H., Peng, S.H., Chen, G.Z. Lee, S.Y. Virtual increase or latent loss? A reassessment of mangrove populations and their conservation in Guangdong, southern China. Marine Pollution Bulletin.2016, 109, 691-699.

[48] Zhang, X.M., Zhou, Y., Liu, P., Wang, F., Liu, B.J., Liu, X.J., Yang, H.S. Temporal pattern in biometrics and nutrient stoichiometry of the intertidal seagrass Zostera japonica and its adaptation to air exposure in a temperate marine lagoon (China): Implications for restoration and management. Marine Pollution Bulletin.2015, 94, 103-113.

[49] Liu, Z.Z., Cui, B.S., He, Q. Shifting paradigms in coastal restoration: Six decades' lessons from China. Science of the Total Environment. 2016, 566-567.

[59] Dempster, T., Arechavala-Lopez, P., Barrett, L.T., Fleming, L.A., Sanchez-Jerez, P., Uglem, I. Recapturing escaped fish from marine aquaculture is largely unsuccessful: alternatives to reduce the number of escapees in the wild. Reviews in Aquaculture.2018, 10,153-167.

[60] Skilbrei, O.T., Wennevik, V. The use of catch statistics to monitor the abundance of escaped farmed Atlantic salmon and rainbow trout in the sea. ICES J. Mar. Sci. 2006, 63, 1190–1200.

[61] Abrantes, K.G., Lyle, J.M., Nichols, P.D., Semmens, J.M. Do exotic salmonids feed on native fauna after escaping from aquaculture cages in Tasmania, Australia? Canadian Journal of Fisheries & Aquatic Sciences.2011, 68, 1539-1551.

[62] Bolstad, G. H., Hindar, K., Robertsen , G., Jonsson, B., Sægrov, H., Diserud, O. H., Fishe, P., Jensen, A.J., Naesje, T.F., Barlaup, B.T., Floro-Larsen, B., Lo, H., Niemela, E., Karlsson, S. Gene flow from domesticated escapes alters the life history of wild Atlantic salmon. Nature Ecology and Evolution.2017, 1, 0124..

[63] Glover, K. A., Solberg, M. F., McGinnity, P., Hindar, K., Verspoor, E., Coulson, M. W., Hansen M.M., Araki, H., Skaala, Y. Svåsand, T. Half a century of genetic interaction between farmed and wild Atlantic salmon: Status of knowledge and unanswered questions. Fish and Fisheries.2017, 18, 890-927.

[64] Hagen, I. J., Jensen, A. J., Bolstad, G. H., Diserud, O. H., Hindar, K., Lo, H., & Karlsson, S. Supplementary stocking selects for domesticated genotypes. Nature Communications. 2019,10, 1-8.

[65] Clavelle, T., Lester, S.E., Gentry, R., Froehlich, H.E. Interactions and management for the future of marine aquaculture and capture fisheries. Fish Fish. 2019, 20,368–388.

[66] Tlusty, M.F., Andrew, J., Baldwin, K., Bradley, T.M. Acoustic conditioning for recall/recapture of escaped Atlantic salmon and rainbow trout. Aquaculture.2008, 274,57-64.

[70] Kitada, S., Nakajima, K., Hamasaki, K., Shishidou, H., Waples, R. S., & Kishino, H. Rigorous monitoring of a large-scale marine stock enhancement program demonstrates the need for comprehensive management of fisheries and nursery habitat. Scientific reports.2019, 9, 5290.

[83] Chai, Z.Y., Huo, Y.Z., He, Q., Huang, X.W., Jiang, X.P., He, P.M, Studies on breeding of Sargassum vachellianum on artificial reefs in Gouqi Island, China. Aquaculture. 2014, 424,189–193.

[91] Wang, W.D., Liang, J., Bi, Y.X., Feng, M.P., Zhou, S.S., Yu, B.C. Status and Prospects of Marine Ranching Construction in Zhejiang Province, Journal of Zhejiang Ocean University (Natural Science).2016,3,181-185.

[103] Rosemond, R.C., Paxton, A.B., Lemoine, H.R., Fegley, S.R., Peterson, C.H. Fish use of reef structures and adjacent sand flats: implications for selecting minimum buffer zones between new artificial reefs and existing reefs. Marine Ecology Progress, 2018, 587,187-199.

[105] Huang, X.Y., Wang, Z.J., Liu, Y., Hu, W.T., Ni, W. On the use of blast furnace slag and steel slag in the preparation of green artificial reef concrete. Constr. Build. Mater.2016, 112, 241–246.

[106] Liu, Y., Zhao, Y.P., Dong, G.H., Guan, C.T., Cui, Y., Xu, T.J. A study of the flow field characteristics around star-shaped artificial reefs. J. Fluids Struct.2013, 39, 27–40.

We would like to show our thanks for your valuable comments, with which we have learned more about marine ranching and artificial reef and improved the quality of our paper.

Special thanks to you again. We wish all goes well with your project.

Other changes:

We tried our best to improve the manuscript and made extensive editing of the English language and style in the manuscript. Meanwhile, we have adapted information and supplement follow editor and reviewer 1’s suggestion as: 

Line 35: change to "traditional"

Line 38: We have used “Therefore, a new research area called the blue growth initiative (BGI) has attracted attention from many fields of science” instead of “Therefore, a new research area called the blue growth initiative (BGI) incorporates technical expertise.”

Lines 45-46 Our original target of this sentence is to connect the BGIs and the next sentence, which “The terms restocking, stock enhancement, culture-based fisheries, capture-based aquaculture, and sea ranching or marine ranching” can be considered as “given set of artificial activities and conditions”. However, since this sentence may cause confusion to readers and we have deleted the sentence. This change will not influence the content and framework

Line 55-65: we have adopting the definition reviewer 1 used in the past “Mustafa defined sea ranching as “releasing juvenile specimens of species of fishery importance raised or reared in hatcheries and nurseries into the sea for subsequent harvest at the adult stage or manipulating fishery habitat to improve growth of the wild stocks [10]”, also ,we adapting two more definition of marine ranching “Grant et al. suggested sea ranching consists of releasing hatchery-reared individuals into the wild, but with the expectation that individuals are harvested before they reproduce or mate with wild individuals [11]; While Kitada point out sea ranching aims at harvesting all juveniles (as possible) released in the harvesting areas, which can be managed by fishers[12];

Lines 69-71: we have used “The major differences in the definition of marine ranching mainly exist between Asian countries and other countries (South America countries, European countries, and Oceanica countries”

Lines 78-79: This sentence may cause confusion to readers and we have deleted the sentence. To compare the difference between different areas, we have used some examples in the paper, see the next point.

Lines 67-78: we have used two compare examples to compare the definition and hot spot of marine ranching between Asian countries and European countries. The comparisons are as following: “For example, in South America countries, European and Oceanica countries, the expectation species for sea ranching are harvested before they reproduce or mate with wild individuals [7,11]. In contrast, Asian countries expected species for sea ranching return as adults to natal rearing areas to spawn [12].”(line 67-73) and “Furthermore, South America countries, Oceanica countries and European countries refer to the use of stock enhancement and restocking strategies (i.e., releasing hatchery seeds to improve or rebuild fishery stocks) [8]. Most cases have shown that beneficiaries of ranching programs fall into four categories: releasing hatchery seeds to improve the self-sustaining populations, releasing hatchery seeds to rebuild severely depleted fish stocks, natural habitat conservation to maintain the habitat function of stock enhancement, and artificial reef construction to create an artificial hard substrate for reef fish [13]. In contrast except focused on developing effective hatchery, release, and field technologies for restocking and commercial wild restock yield management [9], East Asian countries have developed concentrated on a concept of aquaculture-based, artificial habitat-based marine ranching (e.g. buoyant raft and artificial reef) and rehabilitation-based marine ranching(e.g. seaweed bed, seagrass) based on advanced engineering, new materials and structures, which has become a hotspot worldwide since the late 2000s [12,14].

Lines 105-106: we have changed the sentence to “with the goals of reducing capture fishery yield”

Line 147: we have added the reference “[24]”

Line 169-177: we have cut the long sentence into short sentence “The Thirteenth Five-Year Plan declared that, for marine ranching, it would prioritize the regionally representative, high value of ecosystem service functions, scientific management, and social-economic benefits of modern marine ranching in one zone (coastal zone of China) and three regions (Yellow-Bohai Sea, East China Sea, and South China Sea. For marine ranching pilots construction, 178 national marine ranching demonstration areas will be constructed before 2020 (The total construction area is about 1000 km2 area, and 330 km2 of seagrass and seaweed bed will be restored or rehabilitated, and another 50 million stacked cubic meters artificial reef will be released).”

Lines 209-210: we have deleted "of", and change to "targets" in the next line.

Table 1: we have used  "spatial planning based marine ranching" instead of "spatial based marine ranching"

Table 1: we have changed "extinct species" to "regionally extinct species".

Table 1: we have added “public and stakeholder engagement” in “marine ranching management section”

Lines 230: we have changed it to "successfully

Line 251: Fig 1. We have added exactly location of the study area.

Lines 261-265: We have used "However, the evidence suggests that the use of this framework has been successful in several marine ranching settings, so it can be recommended that the steps proposed in Figure 2 may be implemented as proposed to meet the objectives of marine ranching, at least in the ECS." Since the original sentence could be toned down.

Lines 304-309: We have listed some reference in the paper, e.g. “[46-49]”.

Figure 4: we have revised Figure 4.

Line 448: we have changed “restorage” to “restoration”

Line 458-462: we have added a reference “Chai et al. design a new artificial reef with pedestal and breeding broads and a procedure for macroalgae breeding were conducted in Gouqi Island, Zhejiang, China. The speed at which the artificial structures are colonized by plants and animals indicates that artificial habitats have the potential to extend ecosystem functions of natural habitats to offshore areas [83].”

Line 478: we have used “food-chain” instead of “bottom-up cascade”.

Line 480-483: we have revised the sentence as “primary production diversity has strong positive bottom-up effects on secondary producer species (the leakage between primary producers and high trophic level organisms) that graze primary producers directly or extract organic matter for growth”

Line 493-507: we have added two references here [91] and [14]

Line 539: we have provided three references here [83,105,106]

We have used “cultured” here and in line 414, line 553, line 660 and Table 2

Line 621: we have added "the" in front of "same"

Fig.5: we have changed "date" to "data"

Line 653: we have changed “have” to “has”.

Line 664: we have deleted “and numerical” in this sentence.

Line 673: we have changed “its defect is limited” to "its defect is that it is limited"

Line 798: we have changed "judge" to "judgment"

These changes will not influence the content and framework of the paper but improve our paper a lot. And here all the revises are using the "Track Changes" function in Microsoft Word. The number of lines and reference we mentioned is corresponding under the "Track Changes" function.

We appreciate for Editors and Reviewer’s works, suggestions and comments, and hope that the correction will meet with approval.

Once again, thank you very much for your comments and suggestions.

Xijie Zhou, Xu Zhao, Shouyu Zhang, and Jun Lin

Round 2

Reviewer 1 Report

The authors have made satisfactory changes based on the previous comments. I have only a small number of further comments to improve/clarify some of the writing:

line 11-12: the only acronym that is repeated in the abstract is ECS, so I think the others here are unneeded in the abstract and should only be used in the main body text.

line 67: add the end bracket, and change "America" to "American" and "Oceanica" to "Oceanian". Also change those words on line 68 and 71.

line 68-70: change these sentences to "the expectation is that species for sea ranching are harvested before they reproduce or mate with wild individuals [7,11]. In contrast, Asian countries expect that species for sea ranching return as adults to natal rearing areas to spawn"

line 77: change "except" to "besides"

line 163: add an end bracket a the end of this sentence.

line 262: add "are" between "and" and "considered"

line 348: add " the" between "on" and "following"

line 353: change to "salmonids are", and "impact"

line 408: change to "steelhead etc.) that may"

line 422: change "as concerns" to "concerning"

line 444: change "were" to "was"

Author Response

Response to Reviewer 1 Comments

Round 2

Dear Editors and Reviewer 1:

Thank you for your letter and reviewer’s comments concerning our manuscript entitled “Marine Ranching Construction and Management in East China Sea: Programs for Sustainable Fishery and Aquaculture” (ID: water-493706). Those comments are all valuable and very important for revising and improving our paper, as well as guiding significance to our researches. We have studied comments carefully and have made correction which we hope to meet with approval. Revised portions are marked in red in the paper. The main corrections in the paper and the response to the reviewer’s comments are as following:

Point 1: line 11-12: the only acronym that is repeated in the abstract is ECS, so I think the others here are unneeded in the abstract and should only be used in the main body text.

Response 1: We have made correction according to the reviewer’s comments and deleted EAF, EAA and CBA in abstract section. (line 11-12).

Point 2: line 67: add the end bracket, and change "America" to "American" and "Oceanica" to "Oceanian". Also change those words on line 68 and 71.

Response 2: Thank you for reviewer 1's comments, we have added the end bracket, and change “America” to “American” and “Oceanica” to “Oceanian”. Also, we have changed those words on line 69 and line 76.

Point 3: line 68-70: change these sentences to "the expectation is that species for sea ranching are harvested before they reproduce or mate with wild individuals [7,11]. In contrast, Asian countries expect that species for sea ranching return as adults to natal rearing areas to spawn".

Response 3: Thank you for reviewer 2's comments, we have changed these sentences to “the expectation is that species for sea ranching are harvested before they reproduce or mate with wild individuals [7,11]. In contrast, Asian countries expect that species for sea ranching return as adults to natal rearing areas to spawn”. (line 70-73 when using "Track Changes" function)

Point 4: line 77: change "except" to "besides".

Response 4: We have changed “except” to “besides”. (line 82 when using "Track Changes" function)

Point 5: line 163: add an end bracket at the end of this sentence.

Response 5: We have changed this sentence to “social-economic benefits of modern marine ranching in the coastal zone of China, the Yellow-Bohai Sea, the East China Sea, and the South China Sea”. (line 167-168 when using "Track Changes" function)

Point 6: line 262: add "are" between "and" and "considered".

Response 6: We have added "are" between "and" and "considered". (line 269 when using "Track Changes" function)

Point 7: line 348: add "the" between "on" and "following".

Response 7: We have added "the" between "on" and "following". (line 358 when using "Track Changes" function)

Point 8: line 353: change to "salmonids are", and "impact".

Response 7: We have changed “salmonid is” to "salmonids are", and “impactd” to "impact". (line 362 when using "Track Changes" function)

Point 9: line 422: change "as concerns" to "concerning".

Response 9: We have changed "as concerns" to "concerning". (line 431 when using "Track Changes" function)

Point 10: line 444: change "were" to "was".

Response 10: We have changed "were" to "was". (line 454 when using "Track Changes" function)

Special thanks to you for your good comments.

Other changes:

We have tried our best to improve the manuscript and adapted information and supplement follow suggestions and comments of Editor as:

Point 1: The first paragraph of the Introduction should have a quote. Therefore, authors are encouraged to re-write it adding a quote. FAO has abundant references on the subject.

Response 1: Thank you for Editor’s suggestion, we have cited [1], [2], [3] in the first paragraph already, however, because FAO has abundant references on the subject, we have added four additional quotes in the first paragraph to improve the reference citation.

Point 2: Avoid contracted forms (L39, L247, L318, etc).

Response 2: Thank you for suggestion from Editor, we have changed “China’s Agriculture Ministry” to “Ministry of Agriculture of People’s Republic of China” (line 142); “China’s total capture yield” to “total capture yield of China” (line 185); “China’s largest fishing ground” to “largest fishing ground of China” (line 193); “expert’s suggestion” to “expert suggestion” (line 253); “China’s special national conditions” to “special national conditions of China” (line 326); “fisher’s behavior” to “behavior of fisher” (line 800). However, some contracted forms such as “FAO’s BGI”, “People’s Republic of China”, “management’s strategic evaluation (MSE)” and “China’s Thirteenth Five-Year Plan” are the most common forms used in official document and literature, we decide to retain them in the ms.

Point 3: L66-69 must be revised. A bracket is missing and most of “countries” need to be remove for clarity.

Response 3: Thank you for suggestion from Editor, we have added the missing bracket and we have revised the sentence to “The major differences in the definition of marine ranching mainly exist between Asian countries and other countries (South American countries, European countries and Oceanian countries). For example, in South American countries, European countries, and Oceanian countries, the expectation is that species for sea ranching are harvested before they reproduce or mate with wild individuals [7,11]. In contrast, Asian countries expect that species for sea ranching return as adults to natal rearing areas to spawn” follows the suggestion of Reviewer 1. (line 68-73 when using "Track Changes" function)

Point 4: L163-167 must be rephrased, the brackets in the long sentence removed and “pilots”->pilot. 

Response 4: We have rephrased the sentence to “The Thirteenth Five-Year Plan declared that, for marine ranching, it would prioritize the regional representative, high value of ecosystem service functions, scientific management, and social-economic benefits of modern marine ranching in the coastal zone of China, the Yellow-Bohai Sea, the East China Sea, and the South China Sea”. Also, we have changedpilots” to “pilot”. (line 167-169 when using "Track Changes" function)

Point 5: Changes were introduced in Fig 1, but now 2 coordinate systems coexist: decimal degrees and degrees/minutes. This is not acceptable. All coordinates must be provided in decimal degrees. Moreover, an insert with the sub-region must be included for clarity. This will allow the reader to find immediately the location.

Response 5: Thank you for suggestion from Editor.we have provided decimal degrees in Fig 1 and inserted wub-region for clarity the location of marine ranching pilot

Point 6: L443 “Chai et al.” has no quote.

Response 6: We have added a quote in line 452

These changes will not influence the content and framework of the paper but improve our pape. And here all the revises are using the "Track Changes" function in Microsoft Word. The number of lines and reference we mentioned is corresponding under the "Track Changes" function.

We appreciate for Editors and Reviewer’s warm work and hope that the correction will meet with approval.

Once again, thank you very much for your comments and suggestions.

Xijie Zhou, Xu Zhao, Shouyu Zhang, and Jun Lin

2019/05/30
